# THE COUNTING POWER OF TRANSFORMERS

**Marco Sälzer**
RPTU Kaiserslautern-Landau
Kaiserslautern, Germany
`marco.saelzer@rptu.de`

**Chris Köcher**
MPI-SWS
Kaiserslautern, Germany
`ckoecher@mpi-sws.org`

**Alexander Kozachinskiy**
Centro Nacional de Inteligencia Artificial
Santiago, Chile
`alexander.kozachinskyi@cenia.cl`

**Georg Zetzsche**
MPI-SWS
Kaiserslautern, Germany
`georg@mpi-sws.org`

**Anthony Widjaja Lin**
MPI-SWS and RPTU Kaiserslautern-Landau
Kaiserslautern, Germany
`awlin@mpi-sws.org`

## ABSTRACT

Counting properties (e.g. determining whether certain tokens occur more than other tokens in a given input text) have played a significant role in the study of expressiveness of transformers. In this paper, we provide a formal framework for investigating the counting power of transformers. We argue that all existing results demonstrate transformers' expressivity only for (semi-)linear counting properties, i.e., which are expressible as a boolean combination of linear inequalities. Our main result is that transformers can express counting properties that are highly nonlinear. More precisely, we prove that transformers can capture all semialgebraic counting properties, i.e., expressible as a boolean combination of arbitrary multivariate polynomials (of any degree). Among others, these generalize the counting properties that can be captured by C-RASP softmax transformers, which capture only linear counting properties.

To complement this result, we exhibit a natural subclass of (softmax) transformers that completely characterizes semialgebraic counting properties. Through connections with the Hilbert's tenth problem, this expressivity of transformers also yields a new undecidability result for analyzing an extremely simple transformer model — surprisingly with neither positional encodings (i.e. NoPE-transformers) nor masking. We also experimentally validate trainability of such counting properties.

## 1 INTRODUCTION

Transformers (Vaswani et al., 2017) have emerged in recent years as a powerful model with a plethora of successful applications including (among others) natural language processing, computer vision, and speech recognition. Despite the success of transformers, the question of what transformers can express is still not well-understood and has in recent years featured in a rich body of research works (e.g. Strobl et al. (2024); Hahn (2020); Pérez et al. (2021); Hao et al. (2022)). In particular, formal language theory provides a formal framework in understanding expressivity issues for sequential models like transformers and Recurrent Neural Networks (RNNs).

One recurring theme when studying the expressibility of transformers is the *counting power* of transformers. Intuitively, counting amounts to asserting an arithmetic relationship between the numbers of occurrences of various tokens in a given text. Counting properties are essentially the class of properties for textual data under consideration in the well-known *Vector Space Model (VSM)* (cf. Salton et al. (1975); Wong et al. (1985); Shawe-Taylor & Cristianini (2004)), or the similar *Bag-of-Words (BoW)* model (Harris, 1954), which are known from the information retrieval community to be

surprisingly powerful in measuring text similarity (e.g. see Shahmirzadi et al. (2019); Shawe-Taylor & Cristianini (2004)). A simple example of a counting property can be found in a sentiment analysis application[1]: the number of positive words exceeds the number of negative words in a text. In the formal language theory, such a counting property can be formalized as the following language

$$\mathsf{MAJ} := \{w \in \{a, b\}^* : |w|_a > |w|_b\}, \tag{1}$$

which is often referred to as *majority*. Here, $|w|_a$ (resp. $|w|_b$) refers to the number of occurrences of $a$ (resp. $b$) in the string $w$. For example, $\mathtt{aab} \in \mathsf{MAJ}$ but $\mathtt{abb} \notin \mathsf{MAJ}$. Note that "tokens" in NLP are synonymous to "letters" in formal language theory. Another counting property that plays an important role in the theory of expressibility of transformers is *parity* language:

$$\mathsf{PARITY} := \{w \in \{a, b\}^* : a \text{ occurs an even number of times in } w\}. \tag{2}$$

Multiple theoretical and empirical results (e.g. Hahn & Rofin (2024); Chiang & Cholak (2022); Huang et al. (2025); Hahn (2020); Hao et al. (2022); Bhattamishra et al. (2020); Anil et al. (2022); Delétang et al. (2023)) have shown that, while transformers can be efficiently trained for MAJ, this is not the case for PARITY. Several theoretical explanations have been offered, e.g., *sensitivity* by Hahn & Rofin (2024) and length generalization admitted by limit transformers by Huang et al. (2025)).

Thus far, existing results have touched only upon *semilinear* counting properties. For example, defining MAJ requires only a linear inequality (i.e. $|w|_a > |w|_b$). In fact, logical languages, which were devised by Barceló et al. (2024); Yang & Chiang (2024); Huang et al. (2025) epitomizing languages expressible by transformers, permit only linear expressions (e.g. $|w|_a + |w|_b > 2 \cdot |w|_c$). However, polynomial expressions (cf. Shawe-Taylor & Cristianini (2004)) are also used to express *co-occurrence* of terms/tokens in a text. For example, using a *higher-degree* monomial such as

$$\#(\text{nvidia}) \cdot \#(\text{intel}) \cdot \#(\text{deal}),$$

where $\#(w)$ counts the number of occurrences of a word $w$ in the text, one can emphasize the co-occurrence of "nvidia", "intel" and "deal" in a text. This motivates the following question:

**Research Question.** *What counting properties are expressible on transformers? Can they express nonlinear counting properties?*

The main contribution of this paper is the following result.

**Theorem 1.1.** *Transformers can capture all semialgebraic counting properties, i.e., those expressible as a boolean combination of inequalities between multivariate polynomials, where each variable counts the number of occurrences of a specific token in the text.*

This means that transformers can capture expressions involving higher-degree polynomials like $7\#(\text{nvidia}) \cdot \#(\text{intel}) \cdot \#(\text{deal}) + 2\#(\text{shares}) - 8\#(\text{war}) > 10$, or boolean combinations (i.e. unions/intersections) of similar polynomial expressions. Consequently, by the Weierstrass theorem it follows that the set of polynomials can also approximate any continuous function on the number of occurrences of tokens. We prove this theorem (using softmax transformers) — requiring the use of neither positional encodings nor positional masking — and experimentally validate this claim.

Our next question concerns the expressivity of softmax transformers for capturing counting properties: *which class of softmax transformers capture semialgebraic counting properties?* To this end, we provide a surprising characterization involving *average hard attention* (Hao et al., 2022; Pérez et al., 2021), which was devised to "approximate" soft attention by attending to all positions with maximum attention score and forwarding their average. In particular, Average Hard Attention Transformers (AHATs) with only *uniform layers* (written AHAT[U]) — that is, where maximum attention score is achieved at every position — immediately form a subclass of SoftMax Attention Transformers (SMAT). In the sequel, we write NoPE-AHAT (resp. NoPE-AHAT[U]) to mean AHAT (resp. AHAT[U]) that do not use Positional Encodings (PEs) (also no positional masking).

**Theorem 1.2.** NoPE-AHAT *and* NoPE-AHAT[U] *capture precisely semialgebraic counting properties. In particular, as far as expressing counting properties,* NoPE-AHAT *is a subset of* SMAT.

This is surprising, since it is still a major open problem whether AHAT are captured by SMAT (Yang & Chiang, 2024; Hahn, 2020; Yang et al., 2024b) for general (not necessarily counting) properties.

---

[1] https://medium.com/data-science/sentiment-analysis-with-text-mining-13dd2b33de27

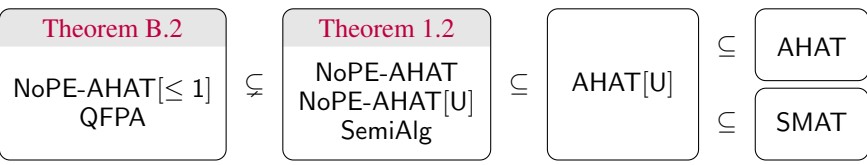

Figure 1: Visualization of our results.

A corollary of Theorem 1.1, combined with Matiyasevich's celebrated solution to the notorious Hilbert's 10th Problem (Matiyasevich, 1993), is a kind of *universality* (i.e. Turing-completeness) of transformers. More precisely, any recursively enumerable counting property $P \subseteq \Sigma^*$ can be represented in terms of a program that, given an input string $w \in \Sigma^*$, feeds each string $wv$ (where $v \in \Gamma^*$, for some $\Gamma \cap \Sigma = \emptyset$) into a transformer $T$ and accepts if $T$ accepts some $wv$. In this case, we say that $P$ is a *projection* of the language accepted by $T$. In fact, we show that transformers $T$ with only two attention layers are sufficient and necessary to achieve this result:

**Theorem 1.3.** *Every recursively enumerable counting property is a projection of a language recognized by a* NoPE-AHAT[U]*, and thus by an* SMAT*. Here, two attention layers in* NoPE-AHAT[U] *and* SMAT *are sufficient.*

Similarly, our results yield an undecidability result for analyzing an extremely simple transformer model—surprisingly with neither positional encodings nor masking:

**Theorem 1.4.** *Given a* NoPE-AHAT[U] *or* SMAT *(with just two attention layers), it is undecidable whether its language is empty.*

Recent results (cf. (Sälzer et al., 2025)) require a substantially more complex architecture to achieve such an undecidability result, i.e., with powerful positional encoding and average hard attention.

Finally, *how do general transformers compare with other machine learning models as far as capturing counting properties?* To this end, let us discuss two models. First is the class of polynomial separators that can be generated by mapping to a higher dimension and look for a linear separator in this higher dimension. This is a standard technique in classical machine learning literature, where one can apply techniques like Support Vector Machines (SVM) (e.g. using polynomial kernel) in the *Vector Space Model (VSM)* (Salton et al. (1975); Wong et al. (1985); also see Chapter 10 of Shawe-Taylor & Cristianini (2004)). Our result shows that transformers generalize such counting properties: not only polynomial counting properties can be captured, but also *boolean combinations* thereof. Second is the model called C-RASP (Huang et al., 2025), which is a simple declarative language that formalizes the so-called *RASP-L conjecture* (Zhou et al., 2024) capturing "efficiently learnable" properties on transformers. In particular, C-RASP allows only linear counting terms. We prove that C-RASP can capture *only* linear counting properties. Since our experiments supporting Theorem 1.1 reveals that counting properties like $L_k := \{w \in \{a, b\}^+ : |w|_a^k \geq |w|_b\}$ are also efficiently learnable for $k \geq 2$, it follows that C-RASP is only a partial characterization of efficiently learnable properties.

**Organization.** We recall transformer models and define our framework for studying counting properties in Section 2. We then show how to capture semialgebraic counting properties using transformers in Section 3. In Section 4, we provide a natural subclass of softmax transformers that completely characterizes semialgebraic counting properties. In Section 5, we show applications of our semialgebraic results for a better understanding of expressiveness of transformers, e.g., universality/undecidability and comparison to work on C-RASP transformers. We report our experimental results in Section 6 and conclude in Section 7. Some details have been relegated into the Appendix.

## 2 FRAMEWORK: TRANSFORMERS AND COUNTING PROPERTIES

**Formal language theory primer** We assume some basic understanding of formal language theory (at the level of a standard undergraduate textbook by Sipser (2013)) and will only fix some notation.

For an alphabet $\Sigma = \{a_1, \ldots, a_m\}$. A *language* is a set of strings over $\Sigma$. We write $\Sigma^*$ (resp. $\Sigma^+$) to mean the set of all strings (resp. all nonempty strings) over $\Sigma$. We write $|w|$ to denote the length of $w$.

For each $a \in \Sigma$, we write $|w|_a$ to mean the number of occurrences of $a$ in $w$. A language $K \subseteq \Sigma^*$ is a *projection* of a language $L \subseteq \Sigma^*$ if there is a subalphabet $\Gamma \subseteq \Sigma$ such that $K$ is obtained from $L$ by deleting all occurrences of letters in $\Gamma$ from words in $L$. For a class $\mathcal{C}$ of languages, by $\mathsf{Proj}(\mathcal{C})$, we denote the class of projections of languages in $\mathcal{C}$.

We will touch upon regular languages and recursively enumerable languages (see Sipser (2013) for details). In summary, regular languages are languages that can be described by regular expressions. *Recursively enumerable* languages are those that are recognized by (possibly nonterminating) Turing machines. The class of such languages is denoted RE. In particular, a machine model is said to be *Turing-complete* if it can capture all recursively enumerable languages.

For an alphabet $\Sigma = \{\mathtt{a}_1, \ldots, \mathtt{a}_m\}$, we define the *Parikh image* (a.k.a. *Parikh map*) as the function $\Psi \colon \Sigma^* \to \mathbb{N}^m$, where $\Psi(w)[i] := |w|_{\mathtt{a}_i}$ is the number of $\mathtt{a}_i$'s in $w$. Intuitively, Parikh image of a word $w$ provides the letter counts in $w$, e.g., over $\Sigma = \{\mathtt{a}, \mathtt{b}\}$, we have $\Psi(abaa) = (3, 1)$. The Parikh map can also be extended to a language $L$; that is, $\Psi(L) = \{\Psi(w) : w \in L\} \subseteq \mathbb{N}^{|\Sigma|}$. For example, if $L = \{\mathtt{a}^n\mathtt{b}^n\mathtt{a}^n : n \geq 0\}$ is a language over $\Sigma = \{\mathtt{a}, \mathtt{b}\}$, we have $\Psi(L) = \{(2n, n) : n \geq 0\}$.

## 2.1 Transformers

We now recall the formal definition of transformers. Loosely speaking, a transformer is a composition of finitely many attention layers, each converting a sequence $\sigma$ of $\mathbb{R}^d$-vectors into another sequence $\sigma'$ of $\mathbb{R}^k$-vectors, for some $d$ and $k$. To turn a transformer $T$ into a language recognizer, we have to embed any letter in the finite alphabet $\Sigma$ as a $\mathbb{R}^d$-vector, where $d$ is smaller than the dimension of the first attention layer. For example, $\Sigma = \{\mathtt{a}, \mathtt{b}, \mathtt{c}\}$, and the *one-hot* embeddings of $\mathtt{a}, \mathtt{b}, \mathtt{c}$ are (respectively) $(1, 0, 0)$, $(0, 1, 0)$, and $(0, 0, 1)$. Finally, to determine acceptance, we simply run $T$ on the embeddings of the input string $w$ into a sequence of vectors (possibly expanded with positional information) and check if the last vector $\boldsymbol{v}$ satisfies that the dot product $\boldsymbol{v}.\boldsymbol{t}$ is greater than 0 (for some pre-defined vector $\boldsymbol{t}$ of weights). In particular, $w$ is accepted by $T$ iff $\boldsymbol{v}.\boldsymbol{t} > 0$.

**Example.** *Suppose we are given the input string $w = \mathtt{abac}$. Additionally, suppose we use the positional embedding $p \colon n \mapsto 1/n$. Then, checking whether $T$ accepts $w$ amounts to running $T$ on the sequence $\sigma$:*

$$(1, 0, 0, 1)(0, 1, 0, 1/2)(1, 0, 0, 1/3)(0, 0, 1, 1/4).$$

*After running $T$ on $\sigma$, the resulting sequence is of the form $\boldsymbol{v}_1, \boldsymbol{v}_2, \boldsymbol{v}_3, \boldsymbol{v}_4$. Determining whether $T$ accepts $w$ amounts to checking whether $\boldsymbol{t}.\boldsymbol{v}_1 > 0$. For example, $\boldsymbol{v}_1, \boldsymbol{v}_2, \boldsymbol{v}_3, \boldsymbol{v}_4$ could be:*

$$(1, 1, 7, 1, 1)(2, 3, 1, 10, 1/2)(1, 8, 0, 8, 1/3)(0, 0, 1, -1, 1/4)$$

*which will be accepted, whenever $t = (1, 0, 0, 1, 0)$.*

Next we formalize the definition of transformers by defining how each attention layer functions.

**ReLU networks.** We first define ReLU networks, which are used inside an attention layer. A *ReLU node* $v$ is a function $\mathbb{Q}^m \to \mathbb{Q}$, where $m \in \mathbb{N}$ is referred to as the input dimension, and is defined as $v(x_1, \ldots, x_m) = \max(0, b + \sum_{i=1}^n w_i x_i)$, where $w_i \in \mathbb{Q}$ are the *weights*, and $b \in \mathbb{Q}$ is the *bias*. [In practice, GeLU and SwiGLU are also used instead of ReLU, which we do not consider in this paper.] A *ReLU layer* $\ell$ is a tuple of ReLU nodes $(v_1, \ldots, v_n)$, all having the same input dimensionality, computing a function $\mathbb{R}^m \to \mathbb{R}^n$, where $n \in \mathbb{N}$ is referred to as the output dimension. Finally, a *ReLU network* $\mathcal{N}$ is a tuple of ReLU layers $(\ell_1, \ldots, \ell_k)$, such that the input dimension of $\ell_{i+1}$ is equal to the output dimension of $\ell_i$. It computes a function $\mathbb{Q}^{m_1} \to \mathbb{Q}^{n_k}$, given by $\mathcal{N}(x_1, \ldots, x_{m_1}) = \ell_k(\cdots \ell_1(x_1, \ldots, x_{m_1}) \cdots)$.

**Attention layers** Each attention layer involves a *weight normalizer* $\mathtt{wt} : \mathbb{R}^* \to \mathbb{R}^*$, which turns any $d$-sequence of weights into another such $d$-sequence. Two widely used weight normalizers are:

1. The softmax normalizer $\mathrm{softmax}$. That is, given a sequence $\sigma = x_1, \ldots, x_n \in \mathbb{R}$, define $\mathrm{softmax}(\sigma) := y_1, \ldots, y_n$, where $y_i := \frac{e^{x_i}}{\sum_{j=1}^n e^{x_j}}$.

2. The averaging hard attention normalizer $\mathrm{aha}$. We define $\mathrm{aha}(\sigma) := y_1, \ldots, y_n$, where

$$y_i := \begin{cases} 1/|P| & \text{if } x_i = \max(\sigma), \\ 0 & \text{or else.} \end{cases}$$

where $P$ consists of positions $i$ in $\sigma$ such that $x_i$ is maximum in $\sigma$. That is, $\mathrm{aha}$ behaves like $\mathrm{softmax}$ but maps all non-maximum weights to 0, and all maximum weights to $1/|P|$.

One can also allow a temperature scaling $\tau > 0$ to $\mathrm{softmax}$, i.e., $\mathrm{softmax}_\tau(\sigma) = y_1, \ldots, y_n$ and set $y_i := \frac{e^{x_i/\tau}}{\sum_{j=1}^n e^{x_j/\tau}}$. This is not so relevant in our paper since our proof works for *any* $\tau > 0$.

An *attention layer* is a function $\lambda \colon (\mathbb{R}^d)^* \to (\mathbb{R}^e)^*$, given by affine maps $Q, K \colon \mathbb{R}^d \to \mathbb{R}^m$, $V \colon \mathbb{R}^d \to \mathbb{R}^k$ (query, key, and value matrices) and a ReLU neural net $\mathcal{N} \colon \mathbb{Q}^{d+k} \to \mathbb{Q}^e$. Given an input sequence $x = (\boldsymbol{x}_1, \ldots, \boldsymbol{x}_n) \in (\mathbb{Q}^d)^n$, the output sequence $y = (\boldsymbol{y}_1, \ldots, \boldsymbol{y}_n) \in (\mathbb{Q}^d)^n$ is computed as follows. First, one computes the sequences of key, query, and value vectors: $\boldsymbol{k}_i = K\boldsymbol{x}_i$, $\boldsymbol{q}_i = Q\boldsymbol{x}_i$, $\boldsymbol{v}_i = V\boldsymbol{x}_i$, for each $i = 1, \ldots, n$, then we define $\boldsymbol{y}_i = \mathcal{N}(\boldsymbol{x}_i, \boldsymbol{a}_i)$, with $\boldsymbol{a}_i = \sum_{j=1}^n \boldsymbol{w}(j)\boldsymbol{v}_j$, where $\boldsymbol{w} = \mathtt{wt}(\{\langle \boldsymbol{k}_i, \boldsymbol{q}_j \rangle\}_{j=1}^n)$.

We say that $\lambda$ is a *softmax* (resp. *aha*) layer if $\mathtt{wt} = \mathrm{softmax}$ (resp. $\mathrm{aha}$). We say that it is a *uniform*-aha layer if it is an $\mathrm{aha}$ layer such that $K\boldsymbol{x} = Q\boldsymbol{x} = \boldsymbol{0}$ for all $\boldsymbol{x}$, i.e., $\langle K\boldsymbol{x}, Q\boldsymbol{y} \rangle = 0$ for all $\boldsymbol{x}$ and $\boldsymbol{y}$. Note that a uniform-aha is both an $\mathrm{aha}$ layer and a $\mathrm{softmax}$ layer since noting that

$$\mathrm{softmax}(s_1, \ldots, s_n) = \mathrm{softmax}_\tau(s_1, \ldots, s_n) = \mathrm{aha}(s_1, \ldots, s_n) = [1/n, \cdots, 1/n],$$

whenever $s_1 = \cdots = s_n$, which can be guaranteed for uniform $\mathrm{aha}$ layers. This holds for *all* $\tau > 0$.

**Remark.** Some papers (e.g. Yang et al. (2024a); Huang et al. (2025); Yang & Chiang (2024)) apply *strict future masking*, which means that attention is only applied to positions up to the current position $i$. Our work does not apply masking.

**Defining transformers.** To define a transformer and its language, we first extend the finite alphabet $\Sigma$ with an *end marker* $\$ \notin \Sigma$. That is, $\Gamma := \Sigma \cup \{\$\}$. A *transformer* with $\ell$ layers over a finite alphabet $\Sigma$ is then a function $T \colon \Sigma^+ \to \{0, 1\}$, given by: (i) the "input embedding" function $\iota \colon \Gamma \to \mathbb{Q}^{d_1}$, (ii) the positional encoding $p \colon \mathbb{N}^2 \to \mathbb{R}^{d_1}$, and (iii) a sequence of layers $\lambda_1 \colon (\mathbb{R}^{d_1})^* \to (\mathbb{R}^{d_2})^*, \ldots, \lambda_\ell \colon (\mathbb{R}^{d_\ell})^* \to (\mathbb{R}^{d_{\ell+1}})^*$. Given an input word $w = a_1 \cdots a_n \in \Sigma^n$, the output $T(w)$ is computed as follows. First, we set $\boldsymbol{x}_1 = \iota(a_1) + p(n+1, 1), \ldots, \boldsymbol{x}_n = \iota(a_n) + p(n+1, n)$, $\boldsymbol{x}_{n+1} = \iota(\$) + p(n+1, n)$. Then we compute $(\boldsymbol{y}_1, \ldots, \boldsymbol{y}_{n+1}) = \lambda_\ell(\lambda_{\ell-1}(\cdots \lambda_1(\boldsymbol{x}_1, \ldots, \boldsymbol{x}_{n+1}) \cdots))$, and we set $T(w) = 1$ if and only if $\boldsymbol{y}_n[1] > 0$, and $T(w) = 0$ otherwise. The language $L(T)$ accepted by $T$ is defined as $\{w \in \Sigma^* : T(w) = 1\}$. We say that $T$ has *no positional encoding* (NoPE) if the positional encoding is a constant function.

**Remark.** Several studies (e.g., Merrill & Sabharwal (2023b); Sälzer et al. (2025); Li & Cotterell (2025)) consider the capabilities of transformers in the context of restricted precision, such as assuming computations are carried out under the assumption of finite representation sizes. We do not focus on these aspects, but note that it is easy to see that our key results, such as Proposition 3.1, also apply under so-called *log-precision* assumptions (cf. Merrill & Sabharwal (2023b); also see Merrill & Sabharwal (2023a)) for rational numbers. This means that the binary representation size of a number $p/q \in \mathbb{Q}$ grows logarithmically with the length of the input.

A *Softmax Attention Transformer* is a transformer using only $\mathrm{softmax}$ layers whereas an *AHA Transformer* is a transformer using only $\mathrm{aha}$ layers. By SMAT we denote the class of all languages accepted by softmax attention transformers and by AHAT we denote the class of all languages accepted by AHA transformers. To all classes we of transformer languages we append "[U]" to denote languages of transformers with only uniform layers, e.g. AHAT[U]. We prepend "NoPE" to denote only languages of transformers with no positional encoding, e.g. NoPE-AHAT[U]. Note that all transformer models we are considering in this paper have only one attention head.

## 2.2 COUNTING PROPERTIES

We now define a framework for studying the counting ability of transformers. Intuitively, our framework focuses on "counting properties". As we shall see below, we can build many interesting formal languages with the help of purely counting properties.

Given a permutation $\pi \colon \{1, \ldots, n\} \to \{1, \ldots, n\}$ and a string $w = w_1 \cdots w_n$ of length $n$, the string $\pi(w) := w_{\pi(1)} \cdots w_{\pi(n)}$ is obtained by permuting the letters in $w$ according to $\pi$.

**Definition 2.1.** A *counting property* over the alphabet $\Sigma$ is a permutation-closed language $L$, i.e., for each $w \in \Sigma^*$, it is the case that $w \in L$ iff $\pi(w) \in L$ for *each* permutation $\pi$ over $\{1, \ldots, |w|\}$.

Examples of counting properties are MAJ and PARITY (see (1), (2)). We often identify a counting property $L$ with its set $\Psi(w) \subseteq \mathbb{N}^{|\Sigma|}$ of letter counts (i.e. Parikh image). By PI, we denote the class of counting properties over $\Sigma$. Counting properties are also called *permutation-invariant* or "proportion-invariant" languages, e.g., see Pérez et al. (2021); Barceló et al. (2024).

**Why counting properties?**   Certainly, many languages of interests have both a "counting component" and an "order component". Take, for example, the language $L_1 = \{\mathtt{a}^n\mathtt{b}^n\mathtt{c}^n : n \geq 0\}$. Our framework focuses on *purely* counting properties for two reasons. Firstly, it abstracts away non-counting components that cannot be captured by the model. Secondly, many formal languages $L$ of interests can be constructed by taking intersection of a counting property $P$ and an order (and counting-insensitive) language $L'$. For example, $L_1$ above can be written as $P \cap L'$, where $P = \{w \in \Sigma^* : |w|_\mathtt{a} = |w|_\mathtt{b} = |w|_\mathtt{c}\}$ and $L' = \mathtt{a}^*\mathtt{b}^*\mathtt{c}^*$. Finally, multiple key languages in the literature on the expressivity of transformers are in fact counting properties (e.g. MAJ and PARITY).

## 3   CAPTURING SEMIALGEBRAIC COUNTING PROPERTIES

A subset $S \subseteq \mathbb{N}^m$ is *semi-algebraic* if it is a Boolean combination of sets of the form $S_p = \{\boldsymbol{x} \in \mathbb{N}^m \mid p(\boldsymbol{x}) > 0\}$ for some polynomial $p \in \mathbb{Z}[X_1, \ldots, X_m]$. A language $L \subseteq \Sigma^*$ is *semi-algebraic* if there is a semi-algebraic set $S \subseteq \mathbb{N}^m$ and $\Sigma = \{\mathtt{a}_1, \ldots, \mathtt{a}_m\}$ such that $L = \{w \in \{\mathtt{a}_1, \ldots, \mathtt{a}_m\}^* \mid \Psi(w) \in S\}$. Let SemiAlg denote the class of semi-algebraic languages. An example is

$$\mathsf{SQRT} = \{w \in \{\mathtt{a}, \mathtt{b}\}^* \mid |w|_\mathtt{a} < |w|/\sqrt{2}\}, , \tag{3}$$

since $|w|_\mathtt{a} < |w|/\sqrt{2}$ if and only if $2|w|_\mathtt{a}^2 < |w|^2$. Likewise, extending the coefficients of our polynomials to rational numbers does not increase the expressiveness of semialgebraic sets, e.g., $\frac{7}{3}xy + y^2 > 8x - 3$ can be rewritten as $7xy + 3y^2 > 24x - 9$. Note that for every $p \in \mathbb{Z}[X_1, \ldots, X_m]$, the set $\{\boldsymbol{x} \in \mathbb{N}^m \mid p(\boldsymbol{x}) = 0\}$ is semi-algebraic, because $p(\boldsymbol{x}) = 0$ if and only if $-p(\boldsymbol{x})^2 + 1 > 0$. Thus, every solution set to polynomial equations is also semi-algebraic.

We show Theorem 1.1. Since AHAT[U] $\subseteq$ SMAT, it sufices to construct a AHAT[U]. We will even construct a NoPE-AHAT[U]. The key ingredient is:

**Proposition 3.1.** *For every polynomial $p \in \mathbb{Z}[X_1, \ldots, X_m]$, the language $L_{p>0} = \{w \in \{\mathtt{a}_1, \ldots, \mathtt{a}_m\}^* \mid p(\Psi(w)) > 0\}$ belongs to* NoPE-AHAT[U]. *Thus, $L_{p>0}$ is in SMAT.*

Let us see why Proposition 3.1 implies SemiAlg $\subseteq$ NoPE-AHAT[U]. First, the complement of each language $L_{p>0}$ can be obtained, because $p(\boldsymbol{x}) > 0$ is violated if and only if $-p(\boldsymbol{x}) + 1 > 0$. Moreover, NoPE-AHAT is closed under union and intersection (we prove a stronger fact in Section A.2). We can thus accept all Boolean combinations of languages of the form $L_{p>0}$, and hence SemiAlg.

To show Proposition 3.1, we will use polynomials that are *homogeneous*, meaning all monomials have the same degree. Note that given an arbitrary polynomial $p \in \mathbb{Z}[X_1, \ldots, X_m]$ of degree $d$, we can consider the polynomial $q \in \mathbb{Z}[X_0, \ldots, X_m]$ with $q = X_0^d p(\frac{X_1}{X_0}, \ldots, \frac{X_m}{X_0})$, which is homogeneous. It has the property that $p(x_1, \ldots, x_m) > 0$ if and only if $q(1, x_1, \ldots, x_m) > 0$. Therefore, from now on, we assume that we have a homogeneous polynomial $q \in \mathbb{Z}[X_0, \ldots, X_m]$ and want to construct an AHAT[U] for the language $K_q = \{w \in \{\mathtt{a}_1, \ldots, \mathtt{a}_m\}^* \mid q(1, \boldsymbol{x}) > 0 \text{ for } \boldsymbol{x} = \Psi(w)\}$.

To simplify notation, we denote the end marker by $\mathtt{a}_0$. Thus, the input will be a string $w \in \{\mathtt{a}_0, \ldots, \mathtt{a}_m\}^+$ that contains $\mathtt{a}_0$ exactly once, at the end. Since $|w|_{\mathtt{a}_0} = 1$ is satisfied automatically, our AHAT[U] only has to check that $q(x_0, \ldots, x_m) > 0$, where $x_i = |w|_{\mathtt{a}_i}$. The input encoding is the map $\{\mathtt{a}_0, \ldots, \mathtt{a}_m\}^* \to \mathbb{Q}^m$ with $\mathtt{a}_i \mapsto \boldsymbol{e}_i$, where $\boldsymbol{e}_i \in \mathbb{Q}^m$ is the $i$-th unit vector.

**Overall idea**   Roughly speaking, we implement multiplication via averaging as follows. For each letter $\mathtt{a}_i$, we have a gadget that can multiply an existing entry $y \in [0, 1]$ (in each vector) by $\frac{x_i}{n+1}$ (recall that $n$ is the overall word length). This is done by first multiplying the existing entries either (i) by 1 if the current letter is $\mathtt{a}_i$ or (ii) by 0 if the current letter is not $\mathtt{a}_i$. This is achieved using a ReLU layer, by observing that for $u \in [0, 1]$ and $v \in \{0, 1\}$, we have $u \cdot v = \mathrm{ReLU}(u - (1 - v))$. After this, we average over the entire input in this component. Since we make sure that all the entries we multiplied with 0 or 1 had the same value $y \in [0, 1]$, taking the average will result in the value $\frac{y \cdot x_i}{n+1}$. Repeating this for a monomial $x_{i_1} \cdots x_{i_d}$, we arrive at the value $\frac{x_{i_1} \cdots x_{i_d}}{(n+1)^d}$. Since our homogenization

step ensured that all our monomials have the same degree $d$, adding up the entries corresponding to the monomials will yield $\frac{p(\Psi(w))}{(n+1)^d}$. Finally, the latter quantity is positive if and only if $p(\Psi(w)) > 0$.

**Step I: Compute frequencies**   Our AHAT[U] first uses an attention layer to compute $m + 1$ new components, where $i$-th component holds $\frac{x_i}{n+1}$, where $n + 1$ is the length of the input (including the end marker). This is easily done by attending to all positions and computing the averages of the first $m + 1$ components. To simplify notation, we will index vectors starting with index 0.

**Step II: Multiplication gadgets**   Second, we have a sequence of gadgets (each consisting of one ReLU layer and one attention layer) that perform the multiplication. Each gadget introduces a new component, and does not change the existing components. Between gadget executions, the following additional invariants are upheld: (i) Overall, a gadget does not change existing components: it introduces one new component. (ii) The components $\{0, \dots, m\}$ are called the *initial* components. (iii) All other components are *uniform*, i.e. they are the same across all positions. (iv) The uniform components carry values in $[0, 1]$. Thus, we will call components $0, \dots, m$ the *initial* components; and we call components $> m$ the *uniform* components.

Our gadgets do the following. Suppose we have already produced $\ell$ additional components. For each initial component $i \in [0, m]$ and uniform component $j \in [m + 1, m + 1 + \ell]$, gadget $\mathsf{omult}(\ell, i, j)$, which introduces a new component, will carry the value $\frac{x_i \cdot y_j}{n+1}$, where $y_j$ is the value in component $j$ of all vectors. Recall that we use $x_i$ to denote the number of $\mathsf{a}_i$ occurrences in the input for $i \in [0, m]$.

We implement the gadget $\mathsf{omult}(\ell, i, j)$ using some ReLU layers and an attention layer. Suppose that before, we have the vector $\boldsymbol{u}_p \in \mathbb{Q}^{m+1+\ell}$ in position $p$. First, using ReLU layers, we introduce a new component that in position $p$ has the value $\boldsymbol{u}_p[i] \cdot \boldsymbol{u}_p[j]$. This can be achieved since $\boldsymbol{u}_p[i]$ is in $\{0, 1\}$ and $\boldsymbol{u}_p[j] \in [0, 1]$: Notice that $\boldsymbol{u}_p[i] \cdot \boldsymbol{u}_p[j] = \mathrm{ReLU}(\boldsymbol{u}_p[j] - (1 - \boldsymbol{u}_p[i]))$. Indeed, if $\boldsymbol{u}_p[i] = 1$, then this evaluates to $\boldsymbol{u}_p[j]$; if $\boldsymbol{u}_p[i] = 0$, then we get $\mathrm{ReLU}(\boldsymbol{u}_p[j] - 1) = 0$. We then use uniform attention to compute the average of this new $\boldsymbol{u}_p[i] \cdot \boldsymbol{u}_p[j]$-component across all vectors. Since there are $n + 1$ vectors, exactly $x_i$ of them have $\boldsymbol{u}_p[i] = 1$, and also $\boldsymbol{u}_p[j] = y_j$, we get the desired $\frac{x_i \cdot y_j}{n+1}$.

**Step III: Computing the polynomial**   We now use our gadgets to compute the value of the polynomial. For each monomial of $q$, say $X_{i_1} \cdots X_{i_d}$, we use $d - 1$ gadgets to compute $x_{i_1} \cdots x_{i_d}/(n+1)^d$: The frequency computation in the beginning yields $x_{i_1}/(n+1)$, and then we use gadgets to compute $x_{i_1} x_{i_2}/(n+1)^2$, $x_{i_1} x_{i_2} x_{i_3}/(n+1)^3$, etc. until $x_{i_1} \cdots x_{i_d}/(n+1)^d$. Finally, we use a ReLU layer to multiply each monomial with a rational coefficient, and compute the sum of all the monomials. Thus, we have computed $q(x_0, \dots, x_m)/(n+1)^d$. We accept if and only if $q(x_0, \dots, x_m)/(n+1)^d > 0$. Note that this is the case if and only if $q(x_0, \dots, x_m) > 0$.

This completes Proposition 3.1 and thus $\mathsf{SemiAlg} \subseteq \mathsf{NoPE\text{-}AHAT}[U]$. We remark that the embedding dimension and the number of layers of our transformer in Proposition 3.1 depends on the degree $d$ and the number $M$ of monomials in $p$. We require at most $O(d)$ layers, each layer increasing the degree of the computed monomials by one. In the appendix, we detailed that polynomials of degree $d$ are accepted by $\mathsf{NoPE\text{-}AHAT}[U]$ using at most $d$ attention layers (see Proposition A.1). The embedding dimension is $O(dM)$ because we store the value of each monomial in a separate dimension.

## 4   CHARACTERIZING SEMI-ALGEBRAIC COUNTING PROPERTIES

We have shown that $\mathsf{NoPE\text{-}AHAT}[U] \subseteq \mathsf{SMAT}$ can capture semi-algebraic counting properties. We now prove that the subclass $\mathsf{NoPE\text{-}AHAT}[U]$ precisely characterizes $\mathsf{SemiAlg}$.

**Proposition 4.1.** $\mathsf{NoPE\text{-}AHAT} \subseteq \mathsf{SemiAlg}$.

*Proof.* Suppose that $\Sigma = \{\mathsf{a}_1, \dots, \mathsf{a}_m\}$ is our alphabet, $\mathsf{a}_0$ the end marker, and $x_i \in \mathbb{N}$ the number of occurrences of $\mathsf{a}_i$ in the input. We say that a position $p$ is an $\mathsf{a}_i$-*position* if the input holds $\mathsf{a}_i$ at position $p$. Notice that an AHAT without positional encoding cannot distinguish vectors that come from the same input letter. This means, in any layer, any two $\mathsf{a}_i$-positions will hold the same vector. Thus, the vector sequence on layer $\ell$ is described by rational vectors $\boldsymbol{u}_{\ell,0}, \dots, \boldsymbol{u}_{\ell,m}$, where $\boldsymbol{u}_{\ell,i}$ is the vector at all the $\mathsf{a}_i$-positions on layer $\ell$. Moreover, for each $i$, the set of positions maximizing an attention score also either contains all $\mathsf{a}_i$-positions, or none of them. Therefore, if the AHAT has $a$ attention layers, there are at most $((2^{m+1})^{m+1})^a = 2^{(m+1)^2 a}$ possible ways to choose the positions of maximal score: On each attention layer, and for each $i \in [0, m]$, we select a subset of the $m + 1$

letters. For each ReLU node and each $i$, there are two ways its expression $\mathrm{ReLU}(v)$ can be evaluated: as 0 or as $v$. Thus, if there are $r$ ReLU nodes, then there are $2^r$ ways to evaluate all those nodes.

For each of these $2^{r+(m+1)^2a}$ choices, we construct a conjunction of polynomial inequalities that verify that (i) this choice actually maximized scores, (ii) the resulting vector at the right-most position in the last layer satisfies the accepting condition. This is easy to do by building, for each layer $\ell$ and each $i$, expressions in $x_1, \ldots, x_m$ for the vectors $\boldsymbol{u}_{\ell,i}$, assuming our choice above. These expressions have the form $p(x_1, \ldots, x_m)/q(x_1, \ldots, x_m)$ (averaging can introduce denominators). Here, once we have expressions for $\boldsymbol{u}_{\ell,i}$, we can use them to build expressions for $\boldsymbol{u}_{\ell+1,i}$ by following the definition of AHAT. Checking (i) and (ii) is then also easy, because inequalities involving quotients $p(x_1, \ldots, x_m)/q(x_1, \ldots, x_m)$ can be turned into polynomial inequalities by multiplying with common denominators. Finally, we take a disjunction over all $2^{r+(m+1)a}$ conjunctions. □

**Inexpressibility of** PARITY. Our characterization of NoPE-AHAT (i.e. Proposition 4.1) implies an interesting inexpressibility result regarding PARITY (see (2)):

**Corollary 4.2.** PARITY *does not belong to* NoPE-AHAT.

PARITY is known to be accepted by AHAT (Barceló et al., 2024) and by SMAT (Chiang & Cholak, 2022) (with PE). Inexpressibility of PARITY in a length-generalizable subclass of SMAT and AHAT (with struct future masking and positional encodings) is known (Huang et al., 2025). Similarly, PARITY is not expressible by SMAT with strict future masking (Hahn, 2020). Corollary 4.2 complements these results and is an easy corollary of Proposition 4.1 (see Section A.3).

## 5 APPLICATIONS

### 5.1 UNIVERSALITY AND UNDECIDABILITY OF TRANSFORMERS

Let us discuss why universality/undecidability (i.e. Theorems 1.3 and 1.4) follow from Theorem 1.2. First, by the well-known theorem "MRDP" theorem (Matiyasevich, 1993) due to Matiyasevich, Robinson, Davis, and Putnam, every language in $\mathsf{RE} \cap \mathsf{PI}$ is a projection of a language of the form $L_p = \{w \in \{\mathtt{a}_1, \ldots, \mathtt{a}_m\}^* \mid p(\Psi(w)) = 0\}$, where $p \in \mathbb{Z}[X_1, \ldots, X_m]$ is a polynomial. Since $L_p$ belongs to NoPE-AHAT[U], we thus obtain Theorem 1.3. Furthermore, since our translation from polynomials to NoPE-AHAT[U] (and thus SMAT) is effective, this also implies Theorem 1.4: By the MRDP theorem (which is also effective), it is undecidable whether a given polynomial $p \in \mathbb{Z}[X_1, \ldots, X_m]$ has a solution. Using our translations, we can turn such a $p$ into a NoPE-AHAT (or SMAT) that is non-empty if and only if $p$ has a solution.

**Using only two layers** In fact, in Theorems 1.3 and 1.4, we even claim that two layers suffice for universality and undecidability. Let us sketch this here. First, our construction above yields a NoPE-AHAT[U] of at most $\ell$ layers, provided that the polynomials in the semialgebraic set all have degree $\leq \ell$ (see Section A). In particular, we show that for each $\ell$, NoPE-AHAT[$\ell$, U] is closed under union and intersection (see Section A.2). Furthermore, we rely on the well-known fact that the set of solutions of a polynomial equation $p = 0$ can always be written as the projection of the set of solutions of a *system of quadratic equations*. Since by our stronger version of Theorem 1.2, intersections of solution sets of quadratic equations only require a NoPE-AHAT[U] with $\leq 2$ layers, this yields the stronger versions of Theorems 1.3 and 1.4. See Section B for details (where we also show that with just one layer, Theorems 1.3 and 1.4 do not hold).

### 5.2 COMPARISON WITH C-RASP AND LTL WITH COUNTING

C-RASP (Huang et al., 2025; Yang & Chiang, 2024) is a simple programming language that can be converted into softmax transformers. In particular, it is a subset of the so-called *LTL with Counting* (Yang & Chiang, 2024; Barceló et al., 2024). For example, $\{w \in \{a, b\}^* : |w|_a = |w|_b\}$ can be written as the following formula in LTL with Counting: $\overrightarrow{\#a} = \overrightarrow{\#b}$. In particular, only linear expressions can be constructed in such formulas. We show in the appendix that LTL with Counting (and therefore C-RASP) only capture (semi)linear counting properties, i.e., boolean combinations of linear inequalities (and modulo arithmetics), so not languages like $L_k := \{w \in \{a, b\} : |w|_a^k \geq |w|_b\}$.

| $k$ | Val. Perf. | Test Perf. | Gen. Perf. |
|---|---|---|---|
| 1 | 0.015 | 0.016/0.99 | 0.301/0.95 |
| 2 | 0.024 | 0.033/0.99 | 0.324/0.94 |
| 3 | 0.023 | 0.021/0.99 | 0.299/0.96 |
| 4 | 0.019 | 0.020/0.99 | 0.099/0.97 |
| 5 | 0.020 | 0.024/0.99 | 0.107/0.96 |

Figure 2: Performance of softmax transformer classifiers for $L_k$ ($k = 1$ to $5$). **Validation Performance (Val. Perf.)**: BCEWithLogitsLoss on validation data. **Test Performance (Test Perf.)**: BCEWithLogitsLoss and Accuracy (separated by /) on test data. **Generalization Performance (Gen. Perf.)**: BCEWithLogitsLoss and Accuracy (separated by /) on generalization test set. The y-axis uses a logarithmic scale to accommodate the different orders of magnitude in the results.

**Proposition 5.1.** *LTL with Counting can define only (semi)linear counting properties.*

## 6 EXPERIMENTS

In this section, we experimentally complement our main result (cf. Theorem 1.1) that transformers can capture solutions of polynomial equations of higher degree. In particular, our results suggest that softmax transformers should be able to learn languages encoding solutions of polynomial equations.

We test our hypothesis on extensions of MAJ with polynomial inequalities. That is, we define the language $L_k$ is defined by $L_k = \{w \in \{a, b\}^+ \mid |w|_b \leq (|w|_a)^k\}$, representing the set of solutions for the simple equation $y \geq x^k$.

> *Do softmax transformer classifiers perform well on language $L_k$? Additionally, can we observe tendencies of length-generalization?*

In other words, the task of the transformer is a binary classification such that $T(w)$ accepts if $w \in L_k$ and it does not if $w \notin L_k$.

We train softmax encoders without positional encoding and otherwise in line with the vanilla model, introduced by Vaswani et al. (2017), as binary classifiers using components offered by Pytorch's `nn.Module` based on a balanced dataset of $5 \cdot 10^5$ data points sampled from $L_k$ for $k = 1, \ldots, 5$ of words up to length 500 In all experiments, we conduct a single epoch and choosed the best model conducting early stopping based on the binary-cross entropy loss combined with softmax, the typical metric for models outputting a probability for binary classification, offered in a numerical stable version by Pytorch's `nn.Module` in form of `BCEWithLogitsLoss`, on a validation dataset sampled from the same distribution and of the same size as the training dataset. To partially explore the hyperparameter space, we conduct a grid search over number of layers 1 to 5, number of heads per layer 1, 2 or 4. In all experiments, we fixed the input features to 32, the feedforward dimension to 64, the dropout rate to 0.3, and optimized using the AdamW optimizer with a learning rate of $10^{-4}$ and weight decay of 0.01 as, again, offered by Pytorch's `optim` package.

Figure 2 presents the outcome of our experiments. The table on the left-hand side demonstrates the best observed performance on the validation dataset (first column), a balanced test dataset derived from the same distribution as the training and validation data (second column). This specifically implies that this dataset also only includes words of length up to 500. The final column represents another balanced test dataset encompassing words from length 501 to 1000, used to potentially unveil some length generalization performance. The plot on the right visualizes the same results.

Generally, we observe very high performance with an accuracy of $\geq 0.99$ on the in-distribution test dataset. Additionally, while the performance on the test dataset with longer words decreases, it remains relatively high, with an accuracy of $\geq 0.94$ in all instances. Especially, it is to be assumed

that with a more extensive experimental setup, this gap in performance will decrease. Therefore, we infer that our trained encoders perform well and that length generalization is supported, indicating that the model can capture the semantics of $L_k$. In Appendix D we report additional results, showing strong performance, with a decrease in performance on longer inputs.

## 7 CONCLUDING REMARKS

**Related Work.** Lots of work have been done in recent years on the expressiveness of transformers for general (not necessarily counting) properties (cf. see (Strobl et al., 2024)). Counting properties — e.g., the languages PARITY and MAJ — have frequently featured in transformers expressivity research, which highlight their importance. Various theoretical transformer models have been used in the literature employing different assumptions on the attention mechanisms (hardmax attention vs. softmax attention), positional encodings, etc. For example, a large proportion of results use hardmax attention, which is not used by practical transformers (which instead use softmax attention). In addition, some works (e.g. Pérez et al. (2021); Barceló et al. (2024)) employ extremely complex positional encodings with no restrictions. That said, several recent works have adopted more practical models. In particular, the works of Yang & Chiang (2024); Huang et al. (2025); Yang et al. (2024b; 2025) employ softmax attention transformers and simple classes of positional encodings (causal masking, local, etc.). Our results also employ a similar model (AHAT[U] and SMAT); in fact, we proved that semialgebraic counting properties can be captured by transformers without any positional encodings. Yang et al. (2025) gave a restriction of softmax attention transformers with bounded finite precision outside the attention computation, which characterizes C-RASP. Our experimental results seem to suggest this transformer model only lower-bounds the expressivity of real-world transformers, which can capture counting properties beyond C-RASP.

Concerning verification of transformers, we mention the works by Yang et al. (2024a) and Bergsträßer et al. (2026), showing that reasoning about Unique-Hard Attention Transformers (UHAT) are decidable with complexity EXPSPACE-complete. UHAT is known to overapproximate what can be captured by softmax transformers with bounded finite precision (Li & Cotterell, 2025). We also mention the recent work (Yang et al., 2026), showing that verifying C-RASP is undecidable.

**Potential Applications in NLP.** By Weierstrass theorem, polynomials can approximate any continuous function of the number of occurrences of tokens. This suggests that transformers can solve practical NLP tasks that require computation of nonlinear statistics in the word frequencies.

Counting properties are tightly connected to *Vector Space Model (VSM)* (Salton et al., 1975; Wong et al., 1985; Shahmirzadi et al., 2019) that has applications in text classification and similarity analysis, where the standard method has been to employ Support Vector Machines (SVM), together with kernel analysis (e.g. using polynomial kernels). Our results imply that transformers are expressive enough to perform such tasks. In VSM, a document $D$ is a vector $v_D$ indexed by "terms" that may occur in $D$. That is, $v_D[t]$ is a count on the number of occurrences of $t$ in $D$. To compare similarity between two documents $D, D'$, we may consider the Euclidean distance between $v_D$ and $v_{D'}$, which requires a polynomial. Also, there are often challenges including "related terms" (e.g. husband, wife, and spouse), which are missed when we only use the aforementioned metric. Thus, a similarity measure is often learned (see Section 10.2.2 in (Shawe-Taylor & Cristianini, 2004), where VSM is used in combination with polynomial kernels). Our results show that transformers can solve such a task. A related task is the problem of determining proximity to a human written text, as dictated by Zipf (1935) stating that the frequency of the $k$-th most frequent word is proportional to $1/k$ in a natural language. As above, we may compare using Euclidean distance a document $D$ with a predetermined Zipf-vector. This results in a polynomial, and our results show this can be captured by transformers.

**Future Work.** We mention several open problems. Firstly, can softmax attention transformers with causal masking capture counting properties beyond semialgebraic sets? Secondly, our work has identified a gap in the formalization of the RASP-L conjecture by Huang et al. (2025). That is, transformers can capture and efficiently learn semialgebraic counting properties, which are beyond the language C-RASP. It is open whether the extension of C-RASP with inequalities over *nonlinear* polynomials can still be captured by softmax transformers.

## ACKNOWLEDGMENTS

We thank David Chiang, Michael Hahn, Andy Yang, and anonymous reviews for their feedback.

Marco Sälzer, Chris Köcher, Georg Zetzsche, and Anthony Lin are funded by the European Union (ERC, LASD, 101089343 and FINABIS, 101077902). Views and opinions expressed are however those of the authors only and do not necessarily reflect those of the European Union or the European Research Council Executive Agency. Neither the European Union nor the granting authority can be held responsible for them.

Alexander Kozachinskiy is funded by the National Center for Artificial Intelligence CENIA FB210017, Basal ANID, and ANID Fondecyt Iniciación grant 11250060.

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

## A   TRANSLATING SEMIALGEBRAIC SETS TO NoPE-AHAT

### A.1   FINE-GRAINED ANALYSIS OF POLYNOMIAL DEGREE VS. DEPTH

In this subsection, we show the inclusion $\mathsf{SemiAlg} \subseteq \mathsf{NoPE\text{-}AHAT}[\mathsf{U}]$. In fact, we show a stronger statement (Proposition A.1), which requires some notation. By $\mathsf{SemiAlg}[\leq \ell]$ we denote the restriction of the class $\mathsf{SemiAlg}$ to the semi-algebraic languages $L \subseteq \Sigma^*$ such that the underlying semi-algebraic set $S \subseteq \mathbb{N}^m$ is a Boolean combination of sets $S_p$ where $p \in \mathbb{Z}[X_1, \ldots, X_m]$ are polynomials of degree $\leq \ell$. In particular, we have $\mathsf{SemiAlg}[\leq 1] = \mathsf{QFPA}$. Our construction for $\mathsf{SemiAlg} \subseteq \mathsf{NoPE\text{-}AHAT}[\mathsf{U}]$ actually shows the following:

**Proposition A.1.** *For each $\ell > 0$ we have* $\mathsf{SemiAlg}[\leq \ell] \subseteq \mathsf{NoPE\text{-}AHAT}[\leq \ell, \mathsf{U}]$.

For showing Proposition A.1, we need some more technical definitions. Let $T$ be an AHAT with input embedding $\iota \colon \Sigma \to \mathbb{Q}^{d_1}$ and layers $\lambda_1 \colon (\mathbb{Q}^{d_1})^* \to (\mathbb{Q}^{d_2})^*, \ldots, \lambda_\ell \colon (\mathbb{Q}^{d_\ell})^* \to (\mathbb{Q}^{d_{\ell+1}})^*$. We define the function $f_T \colon \Sigma^+ \to \mathbb{Q}$ as follows: for a word $w = a_1 a_2 \ldots a_n \in \Sigma^+$, if $\lambda_1 \circ \cdots \circ \lambda_\ell(\iota(a_1), \ldots, \iota(a_n)) = (\boldsymbol{y}_1, \ldots, \boldsymbol{y}_n)$, then $f_T(w) = \boldsymbol{y}_n[1]$. In other words, we have $f_T(w) > 0$ iff $T(w) = 1$.

**Proposition A.2.** *For every polynomial $p \in \mathbb{Z}[X_1, \ldots, X_m]$ of degree $\ell$, the language $L_{p>0} = \{w \in \{a_1, \ldots, a_m\}^* \mid p(\Psi(w)) > 0\}$ belongs to* $\mathsf{NoPE\text{-}AHAT}[\leq \ell, \mathsf{U}]$.

To show Proposition 3.1, we will use polynomials that are *homogeneous*, meaning all monomials have the same degree. Note that given an arbitrary polynomial $p \in \mathbb{Z}[X_1, \ldots, X_m]$ of degree $\ell$, we can consider the polynomial $q \in \mathbb{Z}[X_0, \ldots, X_m]$ with $q = X_0^d p(\frac{X_1}{X_0}, \ldots, \frac{X_m}{X_0})$, which is homogeneous. It has the property that $p(x_1, \ldots, x_m) > 0$ if and only if $q(1, x_1, \ldots, x_m) > 0$. Therefore, from now on, we assume that we have a homogeneous polynomial $q \in \mathbb{Z}[X_0, \ldots, X_m]$ and want to construct an AHAT for the language $K_q = \{w \in \{a_1, \ldots, a_m\}^* \mid q(1, \boldsymbol{x}) > 0 \text{ for } \boldsymbol{x} = \Psi(w)\}$.

To simplify notation, we denote the end marker \$ by $a_0$. Thus, the input will be a string $w \in \{a_0, \ldots, a_m\}^+$ that contains $a_0$ exactly once, at the end. Since $|w|_{a_0} = 1$ is satisfied automatically, our AHAT only has to check that $q(x_0, \ldots, x_m) > 0$, where $x_i = |w|_{a_i}$. The input encoding is the map $\{a_0, \ldots, a_m\}^* \to \mathbb{Q}^m$ with $a_i \mapsto \boldsymbol{e}_i$, where $\boldsymbol{e}_i \in \mathbb{Q}^m$ is the $i$-th unit vector.

In a first lemma we show that each monomial of $q$ can be computed by a NoPE-AHAT with $\ell$ uniform attention layers.

**Lemma A.3.** *For every monomial $r \in \mathbb{Z}[X_0, X_1, \ldots, X_m]$ of degree $\ell$, there is a NoPE-AHAT $T$ with $\ell$ uniform attention layers such that*

$$f_T(w) = \frac{r(\Psi(w))}{|w|^\ell}$$

*for each word $w \in \Sigma^*$. In particular, we have $f_T(w\$) > 0$ if and only if $r(\Psi(w)) > 0$.*

*Proof.* We use the word embedding $\iota \colon \Sigma \to \mathbb{Q}^{m+1}$ with $\iota(a_i) = \boldsymbol{e}_i$ for each $i \in [0, m]$.

**Step I: Compute frequencies** Our AHAT first uses an attention layer to compute $m + 1$ new components, where $i$-th component holds $\frac{x_i}{n+1}$, where $n + 1$ is the length of the input (including the end marker). This is easily done by attending to all positions and computing the averages of the first $m + 1$ components. To simplify notation, we will index vectors starting with index 0.

**Step II: Multiplication gadgets** Second, we have a sequence of gadgets (each consisting of one uniform attention layer and one ReLU layer). Each gadget introduces a new component, and does not change the existing components. Between gadget executions, the following additional invariants are upheld: (i) Overall, a gadget does not change existing components: it introduces one new component. (ii) The components $\{0, \ldots, m\}$ are called the *initial* components. (iii) All other components are *uniform*, i.e. they are the same across all positions. (iv) The uniform components carry values in $[0, 1]$. Thus, we will call components $0, \ldots, m$ the *initial* components; and we call components $> m$ the *uniform* components.

Our gadgets do the following. Suppose we have already produced $k$ additional components. For each initial component $i \in [0, m]$ and uniform component $j \in [m + 1, m + 1 + k]$, gadget $\mathsf{omult}(k, i, j)$, which introduces a new component, will carry the value $\frac{x_i \cdot y_j}{n+1}$, where $y_j$ is the value in component $j$ of all vectors. Recall that we use $x_i$ to denote the number of $a_i$ occurrences in the input for $i \in [0, m]$.

We implement the gadget $\mathsf{omult}(k, i, j)$ using some ReLU layers and an attention layer. Suppose that before, we have the vector $\boldsymbol{u}_p \in \mathbb{Q}^{m+1+k}$ in position $p$. First, using ReLU layers, we introduce a new component that in position $p$ has the value $\boldsymbol{u}_p[i] \cdot \boldsymbol{u}_p[j]$. This can be achieved since $\boldsymbol{u}_p[i]$ is in $\{0, 1\}$ and $\boldsymbol{u}_p[j] \in [0, 1]$: Notice that $\boldsymbol{u}_p[i] \cdot \boldsymbol{u}_p[j] = \mathrm{ReLU}(\boldsymbol{u}_p[j] - (1 - \boldsymbol{u}_p[i]))$. Indeed, if $\boldsymbol{u}_p[i] = 1$, then this evaluates to $\boldsymbol{u}_p[j]$; if $\boldsymbol{u}_p[i] = 0$, then we get $\mathrm{ReLU}(\boldsymbol{u}_p[j] - 1) = 0$. We then use uniform attention to compute the average of this new $\boldsymbol{u}_p[i] \cdot \boldsymbol{u}_p[j]$-component across all vectors. Since there are $n + 1$ vectors, exactly $x_i$ of them have $\boldsymbol{u}_p[i] = 1$, and also $\boldsymbol{u}_p[j] = y_j$, we get the desired $\frac{x_i \cdot y_j}{n+1}$.

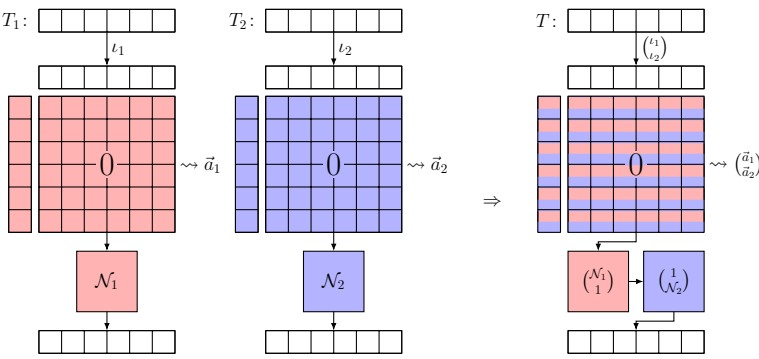

Figure 3: Visualization of the proof of Lemma A.4.

**Step III: Computing the monomial**  We now use our gadgets to compute the value of the monomial. Let $r(X_0, \ldots, X_m) = \alpha \cdot X_{i_1} \cdots X_{i_\ell}$. We use $\ell - 1$ gadgets to compute $x_{i_1} \cdots x_{i_\ell}/(n+1)^\ell$: The frequency computation in the beginning yields $x_{i_1}/(n+1)$, and then we use gadgets to compute $x_{i_1} x_{i_2}/(n+1)^2$, $x_{i_1} x_{i_2} x_{i_3}/(n+1)^3$, etc. until $x_{i_1} \cdots x_{i_\ell}/(n+1)^\ell$. Finally, we use a ReLU layer to multiply $x_{i_1} \cdots x_{i_\ell}/(n+1)^\ell$ with $\alpha$. Thus, we have computed $r(x_0, \ldots, x_m)/(n+1)^\ell$. □

### A.2  COMBINING NoPE-AHAT[U] WITHOUT ADDITIONAL LAYERS

The following lemma states that two NoPE-AHAT with only uniform attention layers can be parallelized resulting in a NoPE-AHAT with the same number of uniform layers. Their outputs can also be combined via a ReLU neural network. In particular, NoPE-AHAT$[\le \ell, \mathsf{U}]$ is closed under union and intersection.

**Lemma A.4.**  *Let $T_1, T_2$ be two NoPE-AHAT with $\ell$ uniform attention layers and let $\mathcal{N}$ be a ReLU neural network computing a function $\mathcal{N}: \mathbb{Q}^2 \to \mathbb{Q}$. Then there is a NoPE-AHAT $T_\mathcal{N}$ with $\ell$ uniform attention layers computing $f_{T_\mathcal{N}}(w\$) = \mathcal{N}(f_{T_1}(w\$), f_{T_2}(w\$))$.*

*Proof.*  The idea of $T_\mathcal{N}$ is, that it concatenates the components from $T_1$ with those of $T_2$ and keeps the sets of components always disjoint. By uniformity we are able to apply the attention layers of $T_1$ and $T_2$ in parallel. In the last attention layer we can simply apply $\mathcal{N}$ to the first components of $T_1$ and $T_2$.

By $\iota_i: \Sigma \to \mathbb{Q}^{d_{1,i}}$ we denote the word embedding of $T_i$. From this we construct a new word embedding $\iota: \Sigma \to \mathbb{Q}^{d_{1,1}+d_{1,2}}$ with $\iota(\mathsf{a}_j) = (\iota_1(\mathsf{a}_j), \iota_2(\mathsf{a}_j))$ for each $j \in [0, m]$.

Now, let $\lambda_{k,i}: \mathbb{Q}^{d_{k,i}} \to \mathbb{Q}^{d_{k+1,i}}$ be the $k$th layer of $T_i$ for $1 \le k \le \ell$. By $K_i, Q_i, V_i$, and $\mathcal{N}_i$ we denote the parameters of $\lambda_{k,i}$. Since $\lambda_{k,i}$ is uniform, the key and query maps $K_i$ and $Q_i$ are constantly mapping to zero. We now construct a uniform layer $\lambda_k: \mathbb{Q}^{d_{k,1}+d_{k,2}} \to \mathbb{Q}^{d_{k+1,1}+d_{k+1},2}$ composed of $\lambda_{k,1}$ and $\lambda_{k,2}$: the key and query maps $K$ and $Q$ still map to zero. If $V_i(\boldsymbol{x}_i) = A_i \boldsymbol{x} + \boldsymbol{b}_i$ then we define the new value map $V$ by

$$V\begin{pmatrix} \boldsymbol{x}_1 \\ \boldsymbol{x}_2 \end{pmatrix} = \begin{pmatrix} A_1 & \mathbf{0} \\ \mathbf{0} & A_2 \end{pmatrix} \begin{pmatrix} \boldsymbol{x}_1 \\ \boldsymbol{x}_2 \end{pmatrix} + \begin{pmatrix} \boldsymbol{b}_1 \\ \boldsymbol{b}_2 \end{pmatrix} = \begin{pmatrix} A_1 \boldsymbol{x}_1 + \boldsymbol{b}_1 \\ A_2 \boldsymbol{x}_2 + \boldsymbol{b}_2 \end{pmatrix} = \begin{pmatrix} V_1(\boldsymbol{x}_1) \\ V_2(\boldsymbol{x}_2) \end{pmatrix}.$$

By this definition we obtain that the attention vectors $\boldsymbol{a}_j$ in $\lambda_k$ are the concatenation of the attention vectors $\boldsymbol{a}_{j,1}$ and $\boldsymbol{a}_{j,2}$ in $\lambda_{k,1}$ resp. $\lambda_{k,2}$. Similarly, we build the composition of $\mathcal{N}_1$ and $\mathcal{N}_2$ resulting in an FFN computing $\binom{\mathcal{N}_1(\boldsymbol{x}_{j,1}, \boldsymbol{a}_{j,1})}{\mathcal{N}_2(\boldsymbol{x}_{j,2}, \boldsymbol{a}_{j,2})}$.

Finally, in the last layer, we add the FFN $\mathcal{N}'$ that takes the first components of the output of $\mathcal{N}_i(\boldsymbol{x}_{j,i}, \boldsymbol{a}_{j,i})$ and simulates $\mathcal{N}$ on these two numbers. □

Recall that from a polynomial $p \in \mathbb{Z}[X_1, \ldots, X_m]$ we constructed a homogeneous polynomial $q \in \mathbb{Z}[X_0, X_1, \ldots, X_m]$ such that $p(\boldsymbol{x}) > 0$ if and only if $q(1, \boldsymbol{x}) > 0$ holds for all vectors $\boldsymbol{x} \in \mathbb{Q}^m$. Let $r_1, \ldots, r_k \in \mathbb{Z}[X_0, X_1, \ldots, X_m]$ be the monomials in $q$. Since $q$ is homogeneous, all

monomials have the same degree $\ell$. Lemma A.3 yields NoPE-AHATs $T_1, \ldots, T_k$ that are computing the monomials $r_i$. Each of these AHATs has exactly $\ell$ uniform attention layers. Finally, we can apply Lemma A.4 to construct a NoPE-AHAT $T$ with $\ell$ uniform layers computing $f_T(w\$) = \frac{q(\Psi(w\$))}{|w\$|^\ell}$ (since addition is an affine map). Then $T$ accepts $w$ iff $\frac{q(\Psi(w\$))}{|w\$|^\ell} > 0$ iff $q(\Psi(w\$)) > 0$ iff $p(\Psi(w)) > 0$. In other words, $T$ accepts the language $L_{p>0}$.

## A.3 INEXPRESSIBILITY OF PARITY

*Proof of Corollary 4.2.* By Theorem 1.2, it suffices to show that PARITY is not semi-algebraic. Suppose it is. Then there is a disjunction of conjunctions of polynomial inequalities that characterizes PARITY. The polynomials are over $\mathbb{Z}[X, Y]$, where $X$ is the variable for $a$'s and $Y$ is the variable for $b$'s. By plugging in $Y = 0$, we conclude that the set of even numbers is semi-algebraic. Hence, there is a disjunction $\bigvee_{i=1}^n \bigwedge_{j=1}^m p_{i,j}(X) > 0$ of conjunctions that is satisfied exactly for the even numbers. This implies that for some $i$, there are infinitely many even numbers $k$ such that $\bigwedge_{j=1}^m p_{i,j}(k) > 0$. Therefore, for every $j \in [1, m]$, the leading coefficient of $p_{i,j}$ must be positive. But then, $\bigwedge_{j=1}^m p_{i,j}(k) > 0$ must hold for all sufficiently large $k$, not just the even ones, a contradiction. $\qquad\square$

## B PARAMETRIC ANALYSIS

In this section, we study how the expressive power of NoPE-AHAT[U] and SMAT depends on the number of attention layers. In particular, we show that Theorems 1.3 and 1.4 hold already in the case of two layers. The main insight of this proof is that the number of layers needed to express a semialgebraic set depends on the degrees of the involved polynomials (see Proposition A.1): Note that our sketch of an NoPE-AHAT for $L_{p>0}$ in Section 3 directly yields a NoPE-AHAT with $\ell$ layers, where $\ell$ is the degree of $p$. For Proposition A.1, one then has to show that Boolean combinations of such sets can be expressed without growing the number of attention layers. See Section A for details.

**Capturing RE with two layers** From Proposition A.1, we can now deduce the two-attention-layer version of Theorems 1.3 and 1.4. The first ingredient is the following version of the MRDP theorem on Diophantine sets (Matiyasevich, 1993):

**Theorem B.1.** *Let $\Sigma = \{a_1, \ldots, a_m\}$. A language $L \subseteq \Sigma^*$ belongs to $\mathsf{RE} \cap \mathsf{PI}$ if and only if there is a $k \in \mathbb{N}$ and a polynomial $p \in \mathbb{Z}[X_1, \ldots, X_{m+k}]$ such that $L = \pi_{a_1, \ldots, a_m}(K)$, where*

$$K = \{w \in \{a_1, \ldots, a_{m+k}\}^* \mid p(\Psi(w)) = 0\}.$$

In other words, every language in $\mathsf{RE} \cap \mathsf{PI}$ is a projection of a language of the form $L_p = \{w \in \{a_1, \ldots, a_m\}^* \mid p(\Psi(w)) = 0\}$, where $p \in \mathbb{Z}[X_1, \ldots, X_m]$ is a polynomial. Thus, it suffices to place $L_p$ in $\mathsf{Proj}(\mathsf{NoPE\text{-}AHAT}[\leq 2, \mathsf{U}])$. First observe that in Theorem 1.2, we use one attention layer for each multiplication, so this avenue is closed if we want to stay within two attention layers. Instead, we use that for every polynomial $p \in \mathbb{Z}[X_1, \ldots, X_m]$, there are *quadratic* (i.e. degree $\leq 2$) polynomials $q_1, \ldots, q_r \in \mathbb{Z}[X_1, \ldots, X_{m+k}]$ for some $r, k \geq 0$ such that for $\boldsymbol{x} \in \mathbb{N}^m$, we have $p(\boldsymbol{x}) = 0$ if and only if there is some $\boldsymbol{y} \in \mathbb{N}^k$ with $q_1(\boldsymbol{x}, \boldsymbol{y}) = 0, \ldots, q_r(\boldsymbol{x}, \boldsymbol{y}) = 0$: Just introduce a fresh variable for each multiplication in $p$ and use the $q_i$ to assign these fresh variables. Since the language $K := \{w \in \{a_1, \ldots, a_{m+k}\}^* \mid q_1(\Psi(w)) = \cdots = q_r(\Psi(w))\}$ belongs to $\mathsf{SemiAlg}[\leq 2]$ (since the $q_i$ have degree $\leq 2$) and $L_p$ is a projection of $K$, this means $L_p$ belongs to $\mathsf{Proj}(\mathsf{SemiAlg}[\leq 2])$. By Proposition A.1, $\mathsf{Proj}(\mathsf{SemiAlg}[\leq 2]) \subseteq \mathsf{Proj}(\mathsf{NoPE\text{-}AHAT}[\leq 2, \mathsf{U}])$.

**NoPE AHAT with a single layer** The fact that two layers suffice for universality among counting properties raises the question of whether this is even possible with a single attention layer. We show here that this is not the case:

**Theorem B.2.** $\mathsf{NoPE\text{-}AHAT}[\leq 1] = \mathsf{NoPE\text{-}AHAT}[\leq 1, \mathsf{U}] = \mathsf{QFPA}$.

This means, with a single attention layer, NoPE-AHAT can recognize precisely those counting properties expressible using quantifier-free Presburger formulas. Since satisfiability of Presburger arithmetic is well-known to be decidable (Haase, 2018; Chistikov, 2024), this implies that universality

and undecidability of NoPE-AHAT (as we have shown for two attention layers), do not hold with just one attention layer. However, we leave open whether SMAT with one attention layer have a decidable emptiness problem.

Before going into details, let us sketch the proof of Theorem B.2. For the inclusion NoPE-AHAT$[\leq 1] \subseteq$ QFPA, we proceed similarly to Proposition 4.1, while observing that the inequalities we have to verify are all linear inequalities: This is because a single attention layer averages only once. Conversely, for the inclusion QFPA $\subseteq$ NoPE-AHAT$[\leq 1, \mathsf{U}]$ follows easily from Proposition A.1.

*Proof of Theorem B.2.* We begin by proving that NoPE-AHAT$[\leq 1] \subseteq$ QFPA. Let $T$ be an AHAT with input embedding $\iota : \Sigma \cup \{\$\} \to \mathbb{Q}^d$, a single AHA layer $\lambda$ utilising affine maps $Q, K \in \mathbb{Q}^{m \times d}$, $V \in \mathbb{Q}^{k \times d}$, given as matrices, and the ReLU network $\mathcal{N} : \mathbb{Q}^{d+k} \to \mathbb{Q}^e$. Our goal is to construct a quantifier-free PA formula $\varphi_T$ with variables $x_i$ for $i \in \{1, \dots, |\Sigma|\}$ such that $\Psi^{-1}(\llbracket \varphi \rrbracket) = \{w \in \Sigma^* \mid T \text{ accepts } w\$\}$. In the following, we assume $\Sigma = \{a_1, \dots, a_m\}$ and denote $\Sigma \cup \{\$\}$ by $\Sigma'$.

First, we observe that for all words $w \in \Sigma^*$, the output of $T$ given $w\$$ is computed by

$$\mathcal{N}\left(\iota(\$), \frac{1}{|w\$|_{a_{i_1}} + \cdots + |w\$|_{a_{i_h}}} \sum_{j=1}^{h} |w\$|_{a_{i_j}} V\iota(a_{i_j})\right),$$

where $\Gamma = \{a_{i_1}, \dots, a_{i_h}\} \subseteq \Sigma'$ is exactly the subset of symbols $a_{i_j}$ occurring in $w\$$ that maximise $\langle Q\iota(a_{i_j}), K\iota(\$)\rangle$. We construct $\varphi_T$ such that it mirrors exactly this computational structure. We have $\varphi_T = \bigvee_{\Gamma \subseteq \Sigma'} \varphi_\Gamma$, where $\bigvee$ ranges over those subsets $\Gamma$ where $\langle Q\iota(a_{i_j}), K\iota(\$)\rangle$ is maximal for precisely the $a_{i_j} \in \Gamma$. The subformula $\varphi_\Gamma$ is defined as follows. For now, we assume that $\$ \notin \Gamma$ and introduce some auxiliary formulas. Throughout the following construction steps, we assume that atomic formulas are normalised to the form $c_1 x_1 + \cdots + c_n x_n \leq b$.

Given the ReLU network $\mathcal{N}$, it is straightforward to construct a quantifier-free PA formula $\varphi^{\mathcal{N}}$ such that $\llbracket \varphi^{\mathcal{N}} \rrbracket$ exactly includes those $x_1, \dots, x_{d+k} \in \mathbb{N}^{d+k}$ satisfying $\mathcal{N}(x_1, \dots, x_{d+k})_1 > 0$, where $\mathcal{N}(\cdot)_1$ denotes the first output dimension of $\mathcal{N}$. The key idea here is that the computation of a single ReLU node $v(x_1, \dots, x_{d+k}) = y$, with weights $c_i$ and bias $b$ of $\mathcal{N}$, is described by the quantifier-free PA formula: $(c_1 x_1 + \cdots + c_{d+k} x_{d+k} + b \leq 0 \wedge 0 = y) \vee (c_1 x_1 + \cdots + c_{d+k} x_{d+k} + b > 0 \wedge c_1 x_1 + \cdots + c_{d+k} x_{d+k} + b = y)$. Then, by nesting this construction iteratively from the last layer to the first layer of $\mathcal{N}$, and finally replacing $= y$ with $> 0$ in the atomic formulas related to the first output dimension of $\mathcal{N}$, we achieve the construction of $\varphi^{\mathcal{N}}$. This nesting and replacement also ensures that $\varphi^{\mathcal{N}}$ includes only the variables $x_1, \dots, x_{d+k}$.

Let $\Gamma \subseteq \Sigma$ such that $\Gamma = \{a_{i_1}, \dots, a_{i_h}\}$. Consider the ReLU network $\mathcal{N}$, the value matrix $V$, and the embedding $\iota$. We construct a quantifier-free PA formula $\varphi_\Gamma^{\mathcal{N}, V}$ such that $\llbracket \varphi_\Gamma^{\mathcal{N}, V} \rrbracket$ exactly includes those $(x_{i_1}, \dots, x_{i_h}) \in \mathbb{N}^h$ satisfying $\mathcal{N}(\iota(\$), \frac{1}{x_{i_1} + \cdots + x_{i_h}} \sum_{j=1}^{h} x_{i_j} V\iota(a_{i_j}))_1 > 0$. To do so, we adjust the formula $\varphi^{\mathcal{N}}$ as described in the following. To account for the fixed input $\iota(\$)$, we replace each occurrence of $x_1$ to $x_d$ in $\varphi^{\mathcal{N}}$ by the respective entry of $\iota(\$)$. Furthermore, to handle the specific form of the input $\frac{1}{x_{i_1} + \cdots + x_{i_h}} \sum_{j=1}^{h} x_{i_j} V\iota(a_{i_j})$, we first replace each occurrence of $x_{d+l}$ with $l \in \{1, \dots, k\}$ in the already modified $\varphi^{\mathcal{N}}$ by:

$$(v_{l1}\iota(a_{i_1})_1 + \cdots + v_{ld}\iota(a_{i_1})_d)x_{i_1} + \cdots + (v_{l1}\iota(a_{i_h})_1 + \cdots + v_{ld}\iota(a_{i_h})_d)x_{i_h},$$

where $v_{lj}$ are the respective entries of $V$. Lastly, we replace each atomic constraint $c_1 x_{i_1} + \cdots + c_h x_{i_h} \leq b$ in the adjusted formula with $(c_1 - b)x_{i_1} + \cdots + (c_h - b)x_{i_h} \leq 0$ to adjust for the factor $\frac{1}{x_{i_1} + \cdots + x_{i_h}}$ present in the input.

Now, we define $\varphi_\Gamma$ as $\varphi_\Gamma^{\mathcal{N}, V, \iota}$. If $\$ \in \Gamma$, we adjust $\varphi_\Gamma^{\mathcal{N}, V, \iota}$ slightly. Assuming $\$ = a_{i_j} \in \Gamma$, we replace the variable $x_{i_j}$ with the constant 1 in $\varphi_\Gamma^{\mathcal{N}, V, \iota}$. Given this construction, it is clear that $\Psi^{-1}(\llbracket \varphi_T \rrbracket) = \{w \in \Sigma^+ \mid T \text{ accepts } w\$\}$, as $\varphi_T$ mimics the computation of $T$ for all possible attention situations $\Gamma$.

For the inclusion QFPA $\subseteq$ NoPE-AHAT$[\leq, \mathsf{U}]$, we observe that QFPA $\subseteq$ SemiAlg$[\leq 1]$, and thus the inclusion follows from Proposition A.1. $\qquad\square$

# C COUNTING PROPERTIES EXPRESSIBLE BY OTHER MODELS

## C.1 SEMILINEAR COUNTING PROPERTIES

A counting property $P \subseteq \mathbb{N}^d$ is said to be *semilinear* if can be defined as a boolean combination of inequalities over linear arithmetic expressions (over variables $x_1, \ldots, x_d$ and integer constants) and modulo arithmetic expressions of the form $x_i \equiv a \pmod{b}$, where $a, b \in \mathbb{N}$ are fixed constants. In particular, semilinear counting properties cannot define semialgebraic counting properties involving polynomials of degrees 2 or above.

It is also convenient to use quantifiers when defining semilinear sets. In particular, they do not increase the expressive power since they can be eliminated. This results in the logic called *Presburger arithmetic (PA)*, which refers to the first-order theory of the structure $\langle \mathbb{N}; +, 0, 1, < \rangle$; see Haase (2018); Chistikov (2024).

## C.2 PERMUTATION-INVARIANT LANGUAGES OF LTL WITH COUNTING

LTL[Count] has the following syntax:

$$\phi ::= a \mid t \leq t \mid \neg \phi \mid \phi \vee \phi \mid X \phi \mid \phi \, U \, \phi$$
$$t ::= k \mid k \cdot \overleftarrow{\#\phi} \mid k \cdot \overrightarrow{\#\phi} \mid t + t$$

where $a \in \Sigma$ and $k \in \mathbb{Z}$. Next we define the semantics of LTL[Count]. For any word $w = a_1 a_2 \cdots a_\ell \in \Sigma^*$ with $a_1, a_2, \ldots, a_\ell \in \Sigma$, for each $1 \leq i \leq \ell$, and each formula $\phi \in$ LTL[Count] we write $w, i \models \phi$ if the formula $\phi$ is satisfied in $w$ at position $i$. Formally, this relation is defined inductively as follows:

- $w, i \models a$ (for $a \in \Sigma$) iff $a_i = a$,
- $w, i \models \neg \phi$ iff $w, i \not\models \phi$,
- $w, i \models \phi \vee \psi$ iff $w, i \models \phi$ or $w, i \models \psi$,
- $w, i \models X \phi$ iff $i < \ell$ and $w, i + 1 \models \phi$,
- $w, i \models \phi \, U \, \psi$ iff there is $i \leq j \leq k$ with $w, j \models \psi$ and for all $i \leq k < j$ we have $w, k \models \phi$,
- $w, i \models t_1 \leq t_2$ iff $[\![t_1]\!](w, i) \leq [\![t_2]\!](w, i)$ where the semantics $[\![t]\!] \colon \Sigma^* \times \mathbb{N} \to \mathbb{Z}$ of a term $t$ is defined as follows: $[\![k]\!](w, i) = k$, $[\![t_1 + t_2]\!](w, i) = [\![t_1]\!](w, i) + [\![t_2]\!](w, i)$, $[\![k \cdot \overleftarrow{\#\phi}]\!] = k \cdot |\{1 \leq j < i \mid w, j \models \phi\}|$, and $[\![k \cdot \overrightarrow{\#\phi}]\!] = k \cdot |\{i \leq j \leq \ell \mid w, j \models \phi\}|$.

Our main result on LTL[Count] is the following:

**Theorem C.1.** *Every permutation-invariant language definable in* LTL[Count] *has a semilinear Parikh image.*

Before we can prove Theorem C.1, we need a few more definitions. For an alphabet $\Sigma$ write $\Sigma_\varepsilon$ for the set $\Sigma \cup \{\varepsilon\}$. A ($d$-dimensional) *Parikh automaton* is a tuple $\mathfrak{A} = (Q, \Sigma, \iota, \Delta, (C_q)_{q \in Q})$ where $Q$ is a finite set of *states*, $\Sigma$ is the *input alphabet*, $\iota \in Q$ is an *initial* state, $\Delta \subseteq Q \times \Sigma_\varepsilon \times \mathbb{N}^d \times Q$ is a finite *transition* relation, and $C_q \subseteq \mathbb{N}^d$ are semilinear *target sets*. A word $w \in \Sigma^*$ is *accepted* by $\mathfrak{A}$ if there are $a_1, a_2, \ldots, a_\ell \in \Sigma_\varepsilon$, states $q_0, q_1, \ldots, q_\ell \in Q$, and vectors $\boldsymbol{v}_0, \boldsymbol{v}_1, \ldots, \boldsymbol{v}_\ell \in \mathbb{N}^d$ such that (i) $q_0 = \iota$ and $\boldsymbol{v}_0 = \boldsymbol{0}$, (ii) for each $0 \leq i < \ell$ there is a transition $(q_i, a_i, \boldsymbol{x}_i, q_{i+1}) \in \Delta$ with $\boldsymbol{v}_{i+1} = \boldsymbol{v}_i + \boldsymbol{x}_i$, and (iii) $\boldsymbol{v}_\ell \in C_{q_\ell}$. The accepted language $L(\mathfrak{A})$ of $\mathfrak{A}$ is the set of all words accepted by $\mathfrak{A}$. It is a well-known fact that for each Parikh automaton $\mathfrak{A}$ the accepted language $L(\mathfrak{A})$ has a semilinear Parikh image. Observe that $0$-dimensional Parikh automata are essentially NFA and, hence, accept exactly the regular languages.

A *Parikh transducer* is a Parikh automaton with input alphabet $\Sigma_\varepsilon \times \Gamma_\varepsilon$ where $\Sigma$ and $\Gamma$ are two alphabets. The accepted language $L(\mathfrak{A}) \subseteq \Sigma^* \times \Gamma^*$ of a Parikh transducer can also be seen as a map: if $(v, w) \in L(\mathfrak{A})$ then we can see $v$ as the input and $w$ as the output of the transducer. Formally, for an input language $L \subseteq \Sigma^*$ a Parikh transducer computes the output $T_{\mathfrak{A}}(L) = \{w \in \Gamma^* \mid \exists v \in L \colon (v, w) \in L(\mathfrak{A})\}$. If $L$ is accepted by a Parikh automaton then $T_{\mathfrak{A}}(L)$ is also accepted by a Parikh automaton. To see this, we can take the synchronized product of the Parikh automaton $\mathfrak{B}$ accepting $L$ and $\mathfrak{A}$ (i.e., $\mathfrak{B}$ reads the same letter from the input as $\mathfrak{A}$ in its first component). Accordingly,

cascading of Parikh transducers is also possible, i.e., if $\mathfrak{A}$ and $\mathfrak{B}$ are Parikh transducers over $\Sigma_\varepsilon \times \Gamma_\varepsilon$ and $\Gamma_\varepsilon \times \Pi_\varepsilon$, we can also construct a Parikh transducer $\mathfrak{C}$ over $\Sigma_\varepsilon \times \Pi_\varepsilon$ computing $T_\mathfrak{C} = T_\mathfrak{B} \circ T_\mathfrak{A}$.

With the definition of Parikh automata and Parikh transducers we are now able to prove Theorem C.1.

*Proof.* Let $\phi \in \mathsf{LTL[Count]}$ be a formula such that the described language $L(\phi)$ is permutation-invariant. We will prove by induction on the structure of $\phi$ that the Parikh image of $L(\phi)$ (or actually a *bounded* subset of $L(\phi)$) is semilinear. Here, a language $L \subseteq \Sigma^*$ is *bounded* if there are letters $a_1, a_2, \ldots, a_n \in \Sigma$ with $L \subseteq a_1^* a_2^* \cdots a_n^*$. So, let $a_1, a_2, \ldots, a_n \in \Sigma$ be distinct letters with $\Sigma = \{a_1, a_2, \ldots, a_n\}$. Then $L(\phi) \cap a_1^* a_2^* \cdots a_n^*$ is clearly bounded and has the same Parikh image as $L(\phi)$.

For each subformula $\psi$ of $\phi$ we construct a Parikh transducer that labels each position satisfying $\psi$. In the base case, we decorate each letter $a$ by $\boldsymbol{b} \in \{0, 1\}^n$ where $\boldsymbol{b}[i] = 1$ iff $a_i = a$. Note that this transducer handles all atomic formulas $a \in \Sigma$ at once. For $\psi = \chi_1 \vee \chi_2$ we add the decoration $b \in \{0, 1\}$ to each letter where $b = 1$ iff one of the decorations corresponding to $\chi_1$ and $\chi_2$ is 1. There are similar transducers (which do not introduce counters) for the cases $\psi = \neg\chi$, $\psi = X\chi$, and $\psi = \chi_1 U \chi_2$. Note that applying these transducers to a bounded language always yields another bounded language.

Now, consider a counting subformula, i.e. $\psi = \sum_{i=1}^{\ell_1} k_i \cdot \overleftarrow{\#\chi_i} + \sum_{i=\ell_1+1}^{\ell_2} k_i \cdot \overrightarrow{\#\chi_i} \leq k$. Observe that the set of positions satisfying $\psi$ is convex in the set of positions satisfying any $\chi_i$. This is true since we consider only a bounded input language. Hence, we can split the input word into three (possibly empty) intervals: (i) the positions at the beginning of the input that do not satisfy $\psi$, (ii) the positions where all positions satisfying a $\chi_i$ also satisfy $\psi$, and (iii) the positions at the end of the input that do not satisfy $\psi$. We describe in the following a Parikh transducer with $3 \cdot \ell_2$ many counters - one for each of these three intervals and each formula $\chi_i$. The transducer guesses the three intervals (note that this is non-deterministic), counts positions satisfying a $\chi_i$ accordingly, decorates only the positions in the second interval labeled with a $\chi_i$ with 1 (and everything else with a 0), and validates in the end our choice of the intervals (via appropriate semilinear target sets ensuring that the equation in $\phi$ is not satisfied in the first and third interval and is satisfied in the second interval). Clearly, this all can be done in one (non-deterministic) Parikh transducer.

Finally, we have a cascade of (Parikh) transducers decorating each position in a bounded input word with a Boolean value indicating whether $\phi$ holds in that position. If we use $a_1^* a_2^* \cdots a_n^*$ as input language for our transducers (note that this language is regular) and intersect the output with all words decorated with a 1 in the first position, we obtain a Parikh automaton accepting exactly the language $L(\phi) \cap a_1^* a_2^* \cdots a_n^*$. Since Parikh automata accept only languages with semilinear Parikh image, we infer that $L(\phi) \cap a_1^* a_2^* \cdots a_n^*$ and, hence, $L(\phi)$ have a semilinear Parikh image. □

## D  FURTHER EXPERIMENTAL VALIDATION

In this section, we report additional experiments addressing a similar research question as posed in Section 6, namely, do softmax transformers perform well on formal languages with inherent non-linear counting properties? Therefore, we consider the language

$$L_{i,j} = \{a^m b^n c^{m^i n^j} \mid m, n \in \mathbb{N}^{\geq 1}\}$$

for selected values of $i$ and $j$. Clearly, recognising this language requires non-linear counting capabilities. Moreover, in contrast to $L_k$ (see Section 6), this language poses a greater challenge in learning tasks due to its structure (all $b$'s follow all $a$'s followed by all $c$'s) and larger alphabet size.

The experimental setup is identical to that presented in Section 6. The results are presented in Figure 4 for five distinct combinations of $i$ and $j$. Similar to our previous experiments, the table on the left shows the highest observed performance on the validation dataset (first column) and the best performance on a balanced test dataset derived from the same distribution as the training and validation data (second column). This indicates that this dataset also contains only words of length up to 500. The final column represents another balanced test dataset of words from length 501 to 1000, utilised to potentially reveal length generalisation performance. The plot on the right visualises the results reported in the table.

| $i,j$ | Val. Perf. | Test Perf. | Gen. Perf. |
|-------|-----------|------------|------------|
| 1,3 | 0.016 | 0.02/0.99 | 0.03/0.99 |
| 3,2 | 0.002 | 0.003/0.99 | 0.60/0.93 |
| 3,3 | 0.001 | 0.002/0.99 | 2.26/0.85 |
| 4,2 | 0.001 | 0.001/0.99 | 0.26/0.96 |
| 5,1 | 0.004 | 0.004/0.99 | 0.03/0.99 |

Figure 4: Performance of softmax transformer classifiers for $L_{i,j}$ (for a selected set of $i$ and $j$ combinations). **Validation Performance (Val. Perf.)**: BCEWithLogitsLoss on validation data. **Test Performance (Test Perf.)**: BCEWithLogitsLoss and Accuracy (separated by /) on test data. **Generalization Performance (Gen. Perf.)**: BCEWithLogitsLoss and Accuracy (separated by /) on generalization test set. The y-axis uses a logarithmic scale to accommodate the different orders of magnitude in the results.

We again observe very high performance of our trained softmax transformers on the in-distribution test dataset (second column), which shares the same distribution as our training dataset. The performance generally remains high on the generalisation test set (third column) as well. We witnessed a slight decrease compared to the results on the in-distribution test in the case of $L_{3,3}$ (accuracy of 0.85). A general decrease in performance on longer inputs is expected and also witnessed in other studies (cf. Huang et al. (2025)), but it also indicates that focused studies are essential to reveal rigorous insights into the relationship between the expressibility of polynomial counting properties we established and their practical learnability.

