# OpenReview forum: "The Counting Power of Transformers"
_ICLR.cc/2026/Conference — ICLR 2026 Poster_

### Official Review · Reviewer_Fyg9 · 2025-10-29

**Soundness:** 3
**Presentation:** 3
**Contribution:** 4
**Rating:** 8
**Confidence:** 4

**Summary:**

This work analyzes the ability of Transformer based models to performing higher order forms of counting (e.g. counting if the product of three variables is larger than some threshold). The authors analyze the counting abilities of Transformers under various assumptions about the self-attention layer and analyze how the counting abilities of models with different self-attention assumptions relate to one another in terms of expressivity.

The main theoretical result of the paper shows that softmax attention transformers can express any counting property which can be expressed as a boolean combination of multivariate polynomial inequalities. For AHAT, the authors provide an exact characterization showing that the set of counting languages accepted by AHAT is equivalent to the set of semi-algebraic languages.

The authors complement this with empirical results showing that models can easily learn to perform classification on the set of languages of the form $|w|_b \leq |w|_a^k$ for k from 1 to 5.

I think the claims made by the paper are very interesting and meaningfully advance the field of Transformer expressivity. Most of my feedback is minor details which I think could help improve the exposition of the theoretical results However, I quite like this paper and thus have given it an 8.

**Strengths:**

- The main claims are interesting and novel. Showing that softmax attention transformers can recognize semi-algebraic counting languages and giving characterization of the subclass meaningfully advances our theoretical understanding of Transformer counting power
- Paper is well written. Motivation of the theoretical problem considered is given. In the sections exposing the framework and the theoretical results, examples are given which is nice.
- At a high level, the theoretical approach taken is clean and convincing. The authors also show an interesting inexpressibility result for PARITY as well as undecidability/universality Theorems (1.3 and 1.4). The argument for two layer networks is also appealing.

**Weaknesses:**

* The main weakness of the paper for me is the exposition. I feel many things could be made clearer or more rigorous in the proofs/in the formalization of the problem. There are many occurrences of variables which are  referred to in text or in equations without properly being defined. I think this hinders the readability of the paper.
* I also think the experiments are quite limited. It would be interested to see evaluation on languages that are not of the form of $L_k$.  I also wish the authors would have tested on languages with different vocabulary sizes to see how this affects performance. However, the theoretical contribution is, in my opinion, strong enough that this is not a major shortcoming to the paper's contribution.

**Questions:**

**General Questions**

- Do we know if, in terms of counting, there are languages in SMAT but **not** in SemiAlg?
- Could the authors give somewhere a more formal/rigorous definition of the sets AHAT, AHAT[U] and SMAT in the context of semi-algebraic language recognition? No formal definition of these sets are given, but many of the main results hinge on inclusions between these and SemiAlg.
- Could you give a more precise theorem statement for Thm 1.1? Which class of Transformers are concerned by this?
- You do not use multi-head attention in your definition of the Transformer. Is there a reason for this?
- I would like for asymptotics (in terms of number of layers/width/number of heads) to be clearly stated either in or around the definition of the theorems. Currently, the only place where number of layers is referred to is in the paragraph discussing the reduction to two layers argument. I found this to be late in the text to introduce these.

**Minor Comments/Clarification Questions**

- For the proof of Prop 3.1"(each consisting of one uniform layer and several ReLU layers)" How many RELU layers? At least state big O.
- The argument for PARITY in the appendix should be introduced in a proof environment with an associated Proposition/Lemma/Theorem.
- What is $d$ line~ 633?
- What are PI and RE languages ~line 382?
- The titles for the columns/lines in Figure 1 could have clearer names.

---

> ### Author Response · Authors · 2025-11-20
>
> Thank you for your comprehensive review.
>
> We address your stated questions below. Feel free to ask follow-up questions.
>
> ---
>
> ### Questions
>
> > Do we know if, in terms of counting, there are languages in SMAT but not in SemiAlg?
>
> The language PARITY is in SMAT (cf. Chiang & Cholak 2022) and not in SemiAlg (cf. Corollary 4.2). But the mentioned transformer is also using positional encoding. To the best of our knowledge, it is not known whether there is also a NoPE-SMAT for PARITY (or any other language not in SemiAlg). We feel that research on the counting capabilities of SMAT is yet in a very early stage and hope to see further results in the future!
>
> > Could the authors give somewhere a more formal/rigorous definition of the sets AHAT, AHAT[U] and SMAT in the context of semi-algebraic language recognition? No formal definition of these sets are given, but many of the main results hinge on inclusions between these and SemiAlg.
>
> The last two paragraphs of Section 2.1 in the original version were meant to define both the language accepted by an individual transformer (the paragraph “Defining Transformers”) and the classes AHAT, AHAT[U], and SMAT of languages accepted by various types of transformers. We made these definitions more explicit. We hope this is clearer than the previous version. Feel free to leave a comment, if the definitions are still unclear.
>
> > Could you give a more precise theorem statement for Thm 1.1? Which class of Transformers are concerned by this?
>
> The precise setting that Thm 1.1 addresses is that of Prop. 3.1., which is discussed in detail in Section 3. To answer your question directly: Thm. 1.1 applies for NoPE-AHAT[U] and SMAT.
>
> > You do not use multi-head attention in your definition of the Transformer. Is there a reason for this?
>
> Thank you for pointing this out! The primary reason is that multi-head attention is not required for any lower bounds we establish. Regarding upper bounds, multi-head attention can be simulated by single-head attention through the addition of layers and increased dimensions of the hidden vectors. Therefore, it does not enhance expressive capabilities. In conclusion, we exclude multi-head attention to maintain simpler definitions and results.
>
> However, if one aims to explore detailed differences in counting capabilities, such as under bounded depth or width assumptions, it would certainly be of interest. We consider this an interesting direction for future research.
>
> > I would like for asymptotics (in terms of number of layers/width/number of heads) to be clearly stated either in or around the definition of the theorems. Currently, the only place where number of layers is referred to is in the paragraph discussing the reduction to two layers argument. I found this to be late in the text to introduce these.
>
> We addressed this in our general response, and added a remark about this at the end of Section 3 in the revised version of the paper.
>
>
> > For the proof of Prop 3.1"(each consisting of one uniform layer and several ReLU layers)" How many RELU layers? At least state big O.
>
> A multiplication gadget consists of one attention layer and one ReLU layer. (We had written “several ReLU layers” in the appendix, because the precise number depends on the exact definition of what a single ReLU layer can do. With our definition of ReLU layers in Section 2.1, a single layer suffices in a multiplication gadget. We have updated this.) The ReLU layer is used to compute the product of $u_p[i]$ and $u_p[j]$, as explained in the paragraph above “Step III”, which shows that we use the single expression $\mathrm{ReLU}(u_p[j]-(1-u_p[i]))$.
>
> > The argument for PARITY in the appendix should be introduced in a proof environment with an associated Proposition/Lemma/Theorem.
> > What is line~ 633?
> > What are PI and RE languages ~line 382?
> > The titles for the columns/lines in Figure 1 could have clearer names.
>
> Thanks for these remarks!
> - We added a proof environment for the result. The formal statement is given in Corollary 4.2.
> - We believe that you are referring to the equation $X_0^d p(...)$? This expression means that we multiply the polynomial $p$ by $X_0$ to the power of $d$, and the statement asserts that the resulting polynomial is still homogeneous.
> - PI is the class of permutation-invariant languages (see l. 255 ff) and RE is the class of recursively-enumerable languages (see l. 158 ff).
> - Thanks for the pointer! We adjusted the figure description and believe that the current style is a good compromise between space and information depicted in the table.

---

> > ### Comment · Reviewer_Fyg9 · 2025-11-24
> > **Official Response by Reviewer Fyg9**
> >
> > Thank you for your thorough response! I believe you have answered all the questions/concerns I had about the paper. I quite like this work and I believe that the theoretical contributions of this work are quite strong. I believe my current score reflects the opinion I have about this paper, thus I will not be changing it.

---

> > > ### Author Response · Authors · 2025-11-28
> > >
> > > Thank you

---

### Official Review · Reviewer_EZ5n · 2025-10-29

**Soundness:** 3
**Presentation:** 2
**Contribution:** 2
**Rating:** 4
**Confidence:** 3

**Summary:**

This paper theoretically investigates the capabilities of encoder-only Transformers to recognize formal languages. Specifically, it focuses on languages characterized by polynomial inequalities based on occurrence counts. The authors leverage this analysis to present a related undecidability result for Transformers. The theoretical findings are empirically validated on the language class $L_k = \{ w\in \{a,b\}^+: |w|_b \leq |w|_a^k \}$.

**Strengths:**

The paper's primary contribution is extending prior work on the counting abilities of Transformers. By generalizing the analysis from linear functions of occurrence counts to polynomial functions, the work advances our theoretical understanding of Transformer capabilities in this domain.

**Weaknesses:**

1. Given the theoretical nature of the paper and the numerous formal language concepts, the presentation would be significantly clarified by adding a Venn diagram. This could visually situate the language classes considered in this work relative to each other and to prior art, making the precise scope of the contribution more apparent.

2. The analysis is confined to encoder-only Transformers. While this is a valid methodological choice, the paper would have broader impact if it discussed the implications of these findings or extended them to autoregressive models, which are prevalent in modern applications.

3. Theorem 1.1 establishes the expressiveness of Transformers for semialgebraic counting properties. However, this claim would be much stronger with a corresponding analysis of the required model size (e.g., depth, width, or number of heads) as a function of the complexity of the counting problem. For instance, do the required Transformers scale polynomially or exponentially with the degree of the polynomial inequalities?

4. The empirical validation of this paper is limited to the single language class $L_k$. The paper's value could be enhanced by either broadening the experimental scope to other complex formal languages or by discussing the potential implications of these counting properties for more conventional NLP tasks, even at a high level.

**Questions:**

1. Could the authors elaborate on the size requirements (e.g., depth, width) for the Transformers needed to express the counting properties discussed in Theorem 1.1 and other results? Specifically, how does the required model size scale with the parameters of the formal language (e.g., the degree of the polynomial)?

2. The current experiments are focused on $L_k$. Are there other language classes the authors considered that would serve as interesting and challenging testbeds for this theory?

3. To help bridge the gap between this theory and practice, could the authors speculate on any practical NLP tasks (e.g., in semantic parsing, logical reasoning, or program synthesis) where these specific polynomial counting abilities might be relevant?

---

> ### Author Response · Authors · 2025-11-20
>
> Thank you for taking the time to do a thorough review!
>
> We address your concerns and questions below.
> To avoid repetition, we will answer parts of your concerns mentioned in "Weaknesses" in the "Questions" part of our comment.
>
> ## Comments on "Weaknesses"
>
> > Given the theoretical nature of the paper and the numerous formal language concepts, the presentation would be significantly clarified by adding a Venn diagram. This could visually situate the language classes considered in this work relative to each other and to prior art, making the precise scope of the contribution more apparent.
>
> Thank you for this great suggestion. We have added a new figure to an updated version of our paper (cf. Fig. 1) in which we compare NoPE-AHAT with the classes AHAT[U], AHAT, and SMAT.
>
> > The analysis is confined to encoder-only Transformers.  [...] extended them to autoregressive models, which are prevalent in modern applications
>
> Thank you for raising this point! We assume that you refer to transformers which "recursively" produce new output tokens (i.e. Chain of Thought (CoT)). When $f(n)$ CoT steps are allowed, hard attention transformers have been shown to capture the set of problems that can be solved by Turing machines that use $f(n)$ steps, e.g., as seen in studies such as
>
> William Merrill, Ashish Sabharwal. "The Expressive Power of Transformers with Chain of Thought" (2023)
> and Pérez et al.'s "Attention is Turing-complete" (2021).
>
> That said, it is still an open problem whether softmax transformers with Chain of Thought are Turing-complete. Our current results entail that softmax transformers are almost Turing complete (see Theorem 1.3). In addition, our results also entail that recognizing solutions to system of polynomial (in)equations can be done *with 0* CoT steps, i.e., very efficiently by transformers. Therefore, we believe that our results serve as a good foundation for attacking the open problem for softmax with CoT (including the fine-grained complexity analysis in terms of the number of CoT steps).
>
>
> ## Questions
>
> > Could the authors elaborate on the size requirements (e.g., depth, width) for the Transformers needed to express the counting properties discussed in Theorem 1.1 and other results? Specifically, how does the required model size scale with the parameters of the formal language (e.g., the degree of the polynomial)?
>
> First, our results require logarithmic precision. Second, the number of attention layers is equal to the degree of the polynomial (a detailed discussion on the number of attention layers can be found in Appendix A.1). We have left a detailed answer about this issue in the general response as the same question was raised by other reviewers as well.
>
> > The current experiments are focused on $L_k$. Are there other language classes the authors considered that would serve as interesting and challenging testbeds for this theory?
> > [...]  could the authors speculate on any practical NLP tasks (e.g., in semantic parsing, logical reasoning, or program synthesis) where these specific polynomial counting abilities might be relevant?
>
> Thanks for the question! As a brief note, to support the rebuttal we added further complementary experiments: we consider languages $L_{i,j}$ over the alphabet $\{a, b, c\}$, where words encode solutions for $x^i \cdot y^j - z = 0$ for different values of $i$ and $j$, in the form $a^x b^y c^{x^i y^j}$. Compared to the languages $L_k$, for which we have already reported experimental results, these $L_{i,j}$ contain a richer structure due to the required order of $a$, $b$, and $c$, and the underlying polynomials have a richer structure than $x^k \geq y$ used for the $L_k$ experiments. For details, see the new Appendix D.
>
> In relation to potential practical NLP applications, we have discussed this in General Response.

---

> > ### Comment · Reviewer_EZ5n · 2025-11-25
> >
> > Thank you for your response.
> >
> > I have a follow-up inquiry regarding the computational complexity. Given that the paper addresses arbitrary polynomials with respect to word frequencies, the current $O(d)$ scaling in depth regarding the degree $d$ appears computationally expensive. Could the authors clarify if it is feasible to reduce the depth, for instance, from $O(d)$ to $O(\log d)$? Although I didn't check the detailed construction, is it possible to apply a divide-and-conquer strategy to optimize the transition from single word frequencies to the final polynomial calculation?

---

> > > ### Author Response · Authors · 2025-11-25
> > >
> > > Thank you for that question - minimizing the number of layers in the resulting
> > > NoPE-AHAT is indeed an intriguing problem!
> > >
> > > However, our proof techniques can be used to show that an $O(\log d)$ upper
> > > bound is not possible: This is because such an upper bound would imply that
> > > there is a constant $D$ such that any semialgebraic set can be defined using
> > > polynomials of degree at most $D$. (Equivalently: a constant number $\ell$ of
> > > layers would suffice to recognize all NoPE-AHAT languages!)
> > >
> > > This follows because our construction in Prop. 4.1 shows that any NoPE-AHAT
> > > with $a$ attention layers can be translated into a semialgebraic set where all
> > > degrees are at most $ra+s$, for some constants $r,s$. If there were a construction
> > > of a NoPE-AHAT with only $c\cdot \log d$ layers for some constant $c$, then
> > > this would mean we can translate any semialgebraic set into a NoPE-AHAT and then
> > > back, and the round-trip translation would yield a degree of just $r\cdot(c\cdot \log d)+s$.
> > > Observe that for some constant $D$, the term $r\cdot (c\cdot \log d)+s$ is strictly
> > > smaller than $d$ for all $d\ge D$. This means, starting with
> > > any semialgebraic set, by applying this round-trip translation again and again,
> > > we can get the degree down to $D$. Thus, any semialgebraic set can be defined
> > > using degree $\le D$.
> > >
> > > However, such a $D$ cannot exist: Consider the semialgebraic set
> > > $P_d=\\{(x,y)\in\mathbb{N}^2 \mid y\le x^d\\}$; we claim that defining $P_d$
> > > requires degree at least $d$. Suppose $P_d$ can be defined using degree $e$.
> > > Then, consider the function
> > > $f\colon\mathbb{N}\to\mathbb{N}$ with $f(x)=\max\\{y\in\mathbb{N} \mid (x,y)\in P_d\\}$.
> > > If $P_d$ can be defined using degree $e$, then we could bound $f$ by a
> > > polynomial of degree $e$: This is because for every choice of $x\in\mathbb{N}$,
> > > the polynomials in the degree-$e$ definition of $P_d$ can be viewed as
> > > univariate polynomials in $y$ with coefficients that are degree-$e$ polynomials
> > > in $x$.  Recall that a univariate polynomial $p$ has a constant sign beyond the
> > > Cauchy bound on its real roots, which is a linear term in the coefficients of
> > > $p$. Hence, all Cauchy bounds of the polyomials in our degree-$e$ definition
> > > are at most $g(x)$, where $g$ is a degree-$e$ polynomial in $x$. But this
> > > implies that $f(x)\le g(x)$.  Therefore, $f(x)=x^d$ is bounded by a degree-$e$
> > > polynomial, and this means $e\ge d$. Therefore, to define $P_d$, we need degree
> > > at least $d$.
> > >
> > > (Note: The above argument rules out not only an $O(\log d)$ upper bound, but
> > > even all upper bounds that grow significantly slower than linear, i.e. $o(n)$).
> > >
> > > Finally: Interestingly, this argument also shows that the number of attention layers of NoPE-AHAT cannot be bounded without losing expressiveness.

---

> > > > ### Comment · Reviewer_EZ5n · 2025-11-27
> > > >
> > > > I appreciate the authors' detailed response, which resolved my initial concerns. Therefore, I would raise my score accordingly.

---

> > > > > ### Author Response · Authors · 2025-11-28
> > > > >
> > > > > Thank you

---

### Official Review · Reviewer_2cZ9 · 2025-10-30

**Soundness:** 4
**Presentation:** 4
**Contribution:** 3
**Rating:** 8
**Confidence:** 4

**Summary:**

The paper provides a theoretical analysis on transformers’ ability to perform counting operations. It is shown that transformers can capture all semialgebraic properties, meaning they can evaluate polynomials where variables are characterized by symbol counts (for instance, accepting strings s.t (#a)^2 + (#b)^3 > 0). The proof mainly consists of demonstrating that average-hard attention transformers (AHATs) with no positional embeddings exactly characterize semialgebraic sets. As a direct corollary, they are able to show that AHATs with no PEs cannot recognize Parity. Furthermore, by connecting this result with the MRDP theorem (which states that there is no algorithm that determines whether some given Diophantine has a solution), they provide an undecidability result for transformers: it is undecidable to determine whether their language is empty or not.

**Strengths:**

1. The perspective on semialgebraic sets rather than semilinear sets is novel and generalizes prior results on the counting power of transformers
2. The corollary on inexpressibility of Parity is interesting and accompanies a rich body of work tackling this question
Novel tools to me such as semialgebraic sets and Parikh images were introduced well enough for me to understand the technical parts of the paper
3. Introduction is well written and puts in perspective previous work on semilinear counting with transformers

**Weaknesses:**

1. Importantly, this paper disregards the impact of precision. In the finite-precision regime (which describes transformers used in practice), it is impossible to store counts from uniform attention for any input string. Recently, the expressive power of fixed-precision transformers (SMATs and AHATs) has already been characterized by a subclass of regular languages [Li and Cotterell, 2025] (and therefore can not perform counts across all possible strings), undermining the relevance of the paper’s main claim on counting. I would recommend at least mentioning this inherent limitation.
2. The experiments consist of training transformers on a single type of polynomial (comparing (#b) and (#a)^k) rather than a more diverse set of polynomials with different coefficients of different degrees. Experiments on polynomials with a larger alphabet, different coefficients with different degrees would consolidate the claims of the paper. Even better, training on randomly sampled polynomials would be a great contribution.

**Questions:**

N/A

---

> ### Author Response · Authors · 2025-11-20
>
> Thank you for your review!
>
> We comment on the weaknesses you mentioned below.
> If you have any additional questions, feel free to discuss with us.
>
> ## Comments on "Weaknesses"
>
> > Importantly, this paper disregards the impact of precision. In the finite-precision regime (which describes transformers used in practice), it is impossible to store counts from uniform attention for any input string. Recently, the expressive power of fixed-precision transformers (SMATs and AHATs) has already been characterized by a subclass of regular languages [Li and Cotterell, 2025] (and therefore can not perform counts across all possible strings), undermining the relevance of the paper’s main claim on counting. I would recommend at least mentioning this inherent limitation.
>
> The issue of precision is indeed important. Our construction of a transformer for semialgebraic languages needs just logarithmic precision, please see the general response to all reviewers. We also are grateful for the reference to [Li and Cotterell, 2025]. We added a remark about the assumptions of precision we adopted and referenced this paper in Section 2.1.
>
> That being said, we would like to argue that logarithmic precision assumption, required for our results, might be as relevant for practical transformers as the fixed-precision one. On the one hand, yes, numbers in computers are stored with some fixed number of bits. On the other hand, transformers are not run on arbitrarily large inputs with this precision. Hence, it becomes a question of which asymptotic relationship between the number of precision bits and input lengths more adequately represents practical transformers. One might note that the standard precision length in Python, 64 bits, is more than enough to cover most practical input lengths to transformers ($2^{64}$ is approx. $2\cdot 10^{19}$). In this range, practical transformers do have an ability to perform counting via uniform attention. In particular, transformers have been shown to achieve near perfect accuracy in computing the majority function which is not regular, see [Butoi et al., Training neural networks as recognizers of formal languages (2024) and also Huang et al. (2025)].
>
> > The experiments consist of training transformers on a single type of polynomial (comparing (#b) and (#a)^k) rather than a more diverse set of polynomials with different coefficients of different degrees. Experiments on polynomials with a larger alphabet, different coefficients with different degrees would consolidate the claims of the paper. Even better, training on randomly sampled polynomials would be a great contribution.
>
> Thank you for the suggestion! As you noted in your review, the primary contribution of our paper is theoretical, which is why our complementary experimental validations may appear less extensive in comparison.
>
>
> However, we address your concern as follows:
> - We have clarified the description of Figure 1 and extended the experimental description, which was previously in Appendix D, and moved it to the main body of the paper. This ensures the experimental section is self-contained and should improve the understanding that our experimental setting is limited and only complementary to our theoretical contribution.
> - Additionally, we have added further experiments on languages $L_{i,j}$ over the alphabet $\{a,b,c\}$, where words encode solutions for $x^i \cdot y^j - z=0$ for different values of $i$ and $j$, in the form $a^x b^y c^{x^i y^j}$. Compared to the languages $L_k$, for which we have already reported experimental results, these $L_{i,j}$ are (a) non-binary languages, (b) contain a richer structure due to the required order of $a$, $b$, and $c$, and (c) the underlying polynomials have a richer structure than $x^k \geq y$ used for the $L_k$ experiments. The results of these experiments underline what we found before: transformers perform well if we test in-distribution and show some length generalisation tendencies, but dedicated studies are needed to thoroughly unveil the interplay of theoretical expressiveness results such as ours and practical learnability. Please, see the new Appendix D of the revised version for this.

---

> > ### Comment · Reviewer_2cZ9 · 2025-11-21
> >
> > Thank you for your response. I maintain my positive score.
> >
> > I appreciate your comments about finite vs. log-precision transformers. Your reasoning in favor of log precision makes sense. In fact, it is quite similar to Section 3 here: https://arxiv.org/pdf/2210.02671

---

> > > ### Author Response · Authors · 2025-11-28
> > >
> > > Thank you for pointing out the paper. We've added a pointer to the paper in the new version.

---

### Official Review · Reviewer_XAw7 · 2025-11-01

**Soundness:** 3
**Presentation:** 3
**Contribution:** 2
**Rating:** 4
**Confidence:** 4

**Summary:**

This paper investigates the counting power of Transformers.
While prior work on Transformer expressiveness has shown that Transformers possess semilinear counting properties, this paper demonstrates that Transformers can express all semialgebraic counting properties.
It also shows that Average Hard Attention Transformers (AHAT) without positional encodings (PEs), as well as their subset AHAT[U] that uses only uniform layers, precisely capture semialgebraic counting properties.

**Strengths:**

- Understanding the expressive power of Transformers is an important research topic.
  This paper presents new results on counting capabilities that were not clarified in previous work.
- The paper is concise and readable.
- The theoretical results are supported by experiments.
- Although I had some questions, the theoretical part appears mostly correct.

**Weaknesses:**

- It is unclear how practical it is to apply Transformers to nonlinear counting problems as discussed in this paper.
  For sequences such as text, which Transformers typically handle, input order is important, and tasks are generally not permutation-invariant.
  Therefore, studying permutation-invariant input properties may have limited practical relevance.
  As the authors discuss in the paragraph beginning at Line 263, combining counting properties with other characteristics is interesting.
  However, it is not clear whether nonlinear counting is necessary in such cases.
  In fact, the use case shown at Line 264 can be realized with linear counting.

- The proof of the main result, Proposition 3.1, seems somewhat trivial.
  The idea in Step I, computing the frequency via a Transformer layer, has already appeared in prior work such as Yang & Chiang (2024).
  Step II, which performs multiplication, also appears straightforward.


## Minor comments

- Line 22: properties.. => properties.
- Line 161: (a.k.a. Parikh map => missing ')'
- Line 180: I think the fourth vector should be (0,0,1, 1/4) if we use the one-hot embeddings.
- Line 190: (x_1, \ldots, x_n) => (x_1, \ldots, x_m)?
- Line 205: duplicated ':='.
- Line 213: x_j \tau => x_j / \tau
- Line 227: duplicated ','.

**Questions:**

- Are there practical situations where nonlinear counting is required for tasks usually solved with Transformers?
- Is it possible to define a new class of language, e.g., LTL with nonlinear counting?
- I could not understand the paragraph beginning at Line 355.
  Regarding the arbitrary choices among the $2^{r+(m+1)^2a}$ options representable by AHAT, why is it “easy to do” to construct the polynomial expression of $u\_{\ell, i}$?

---

> ### Author Response · Authors · 2025-11-20
> **Comments on Weaknesses**
>
> Thanks for taking time to do a comprehensive review of our paper. We address all of your questions and comments in detail below.
>
> ---
>
> ### Comments on "Weaknesses"
>
> > It is unclear how practical it is to apply Transformers to nonlinear counting problems as discussed in this paper.
>
> See General Response.
>
> > For sequences such as text, which Transformers typically handle, input order is important, and tasks are generally not permutation-invariant.
> Therefore, studying permutation-invariant input properties may have limited practical relevance.
>
> We do agree that the word ordering plays an important role in most Transformer applications. However, we would like to pinpoint that pure permutation-invariant properties are certainly not irrelevant. As we explained above, VSM is a classic and practically relevant technique that has been rather successful in the information retrieval community (e.g. for detecting text similarity), which relies purely on permutation-invariant properties. See the cited book by Shawe-Taylor and Cristianini (2004) and:
>
> https://arxiv.org/pdf/1810.00664
>
> Finally, if we want to understand the expressive power of transformers in general, counting properties provide a clean separation of concerns and abstraction that is more amenable to theoretical analysis and can be composed with non-permutation-invariant properties (e.g. under taking intersection). We believe that our results constitute an important milestone towards solving a problem that you mentioned regarding LTL with nonlinear counting (see below).
>
> > As the authors discuss in the paragraph beginning at Line 263, combining counting properties with other characteristics is interesting.
> However, it is not clear whether nonlinear counting is necessary in such cases.
> In fact, the use case shown at Line 264 can be realized with linear counting.
>
> We do agree that it is interesting to study languages that combine counting properties and a non-permutation invariant feature. The language on Line 264 (in the submitted version) can indeed be recognized with only linear counting. Using similar reasoning, our results also entail that the following languages can also be recognized by transformers:
>
> $L_{i,j} = \\{a^m b^n c^{m^i n^j} \mid m,n \in \mathbb{N}^{\geq 1}\\}.$
>
> We have done additional experiments (see Section 6 and Appendix D) confirming our theoretical prediction that this language can also be learnt by transformers.
>
> > The proof of the main result, Proposition 3.1, seems somewhat trivial. The idea in Step I, computing the frequency via a Transformer layer, has already appeared in prior work such as Yang & Chiang (2024). Step II, which performs multiplication, also appears straightforward.
>
> We acknowledge there are similarities between Step 1 of our proof of Prop. 3.1 and some of the arguments used in the study by Yang & Chiang (2024) you mentioned, even though they generally consider different settings like masked transformers. This step can be very roughly described as “building an attention layer that attends to all positions”, or in other words, using a uniform attention layer, which is in itself not much of a challenge.
>
> However, we respectfully disagree that Prop. 3.1 is straightforward or not a worthy contribution. We believe our results should generally be viewed in the broader context of improving our understanding of the counting capabilities of transformers, which Prop. 3.1, as the key argument for Theorem 1.1, does. Steps 2 and 3 involve careful constructions to (a) keep the averaging behaviour of AHAT[U]/SMAT layers under control, and (b) ensure that the potentially encoded solution for the polynomial $p$ can be correctly decoded in the end.
>
> For example, while averaging over all vectors is standard, we use this to implement multiplication: To multiply by $x_i$, we first multiply all entries carrying $a_i$ by 1, and multiply all others by 0. It is key that this particular multiplication can be done using ReLU layers. After this, averaging all positions yields precisely the product with $x_i$, but divided by $(n+1)$. To mitigate these undesirable $(n+1)$ factors, we first make our polynomial homogeneous: This has the effect that every monomial receives the same product $(n+1)^d$ in its denominator. Since all monomials receive the same factor, we can then observe that it does not affect negativity of the entire expression. In our view, these tricks are not obvious. In fact, it is unclear if the result of Yang & Chiang (2024) extends to non-linear counting — or equivalently extending our setting to masked transformers  — which we have mentioned in Section 7 as an open problem.
>
> > Minor comments
>
> Thanks for pointing these out. We have fixed them in the revised version of the paper.

---

> ### Author Response · Authors · 2025-11-20
> **Comments on Questions**
>
> > Are there practical situations where nonlinear counting is required for tasks usually solved with Transformers?
>
> Thanks for raising this question. We responded to it in our comment of the related weakness you pointed out above.
>
> > Is it possible to define a new class of language, e.g., LTL with nonlinear counting?
>
> Defining is certainly possible, however it is completely unclear what its connection would be to transformers (as we already remarked above). In particular, if we extend C-RASP of Yang and Chiang (2024) with nonlinear counting, we do not know anymore if this is still expressible in SMAT or even AHAT (even when allowing non-uniform layers). We believe that our results constitute an important milestone in answering this open question. We have also left this for future work in Section 7.
>
> > I could not understand the paragraph beginning at Line 355. Regarding the arbitrary choices among the $2^{r+(m+1)^2a}$ options representable by AHAT, why is it “easy to do” to construct the polynomial expression of $u_{l,i}$?
>
> Here is a more detailed explanation: Let $x_0, x_1, ..., x_m$ be the numbers of occurrences of letters ($x_0 = 1$ as this corresponds to the unique end marker). Assume that we already have computed $u_{l, 0},u_{l,1}, …, u_{l, m}$. Now, how does one compute $u_{l+1,0},u_{l+1,1}, …, u_{l+1, m}$? For each $i \in \\{0, .., m\\}$, we have a set $S_i \subseteq \\{0, 1, …, m\\}$, describing the set of all $j$ with the maximal attention from $a_i$-positions to $a_j$-positions. To compute, $u_{l+1, i}$, we need to compute the average of values of transformers in $a_j$-positions over $j \in S_i$. There are $\sum_{j \in S_i} x_j$ such positions. Now, the sum of values in these positions becomes $\sum_{j \in S_i} u_{l,j} x_j$. The overall expression for the corresponding language becomes:
>
> $(\sum_{j \in S_i} u_{l,j} x_j) / (\sum_{j \in S_i} x_j )$
>
> which is a rational function in $x_1, …, x_m$ (the fraction of two polynomials) if $u_{l,j}$ are rational functions. We can turn them into polynomial expressions inside our system of inequalities by multiplying by common denominators.

---

> > ### Comment · Reviewer_XAw7 · 2025-11-21
> >
> > Thanks for the response. As explained in the general response, I agree that the paper’s demonstration that Transformers can solve tasks corresponding to vector space models is an important contribution. I now also understand that the proofs in the paper are not straightforward. Overall, I believe this work enhances our understanding of the expressivity of Transformers, and I will raise my score accordingly.

---

> > > ### Author Response · Authors · 2025-11-27
> > >
> > > Thank you. We very much appreciate it and will be happy to answer further questions.

---

### Official Review · Reviewer_sqAV · 2025-11-05

**Soundness:** 3
**Presentation:** 4
**Contribution:** 4
**Rating:** 8
**Confidence:** 4

**Summary:**

This paper formally studies the counting power of transformers, i.e., which counting properties of the input can transformer models represent. The authors substantially expand on prior understanding, showing that the answer goes well-beyond previously studied (Boolean combinations of) linear properties, to (Boolean combinations of) polynomial properties of the number of occurrences of various tokens in the input. Along the way, they also define and motivate counting properties, and provide a small empirical confirmation of the findings. Towards the end, they make connections to universality and undecidability of certain classes of transformers.

**Strengths:**

* Well-motivated and nicely written paper. While technically strong, its pitch and claims will also be accessible to broader audience not deep into theoretical research.

* I liked the structure of the paper and how it (quickly) conveys the findings and uses many examples to explain concepts.

* That the studied transformer model uses simple (or no) position encodings and the standard softmax attention (either directly or through AHAT[U] which is a special case of softmax) is a positive, especially compared to some efforts that have relied to unrealistically complex position embeddings and other design choices in their analysis.

* The main result about transformers being able to capture counting properties well beyond (combinations of) linear properties is a strong technical advance. Having an empirical support for it is a plus.

* The connections to undecidability are intriguing, though I must say the corresponding sections can use some more clarity of exposition.

**Weaknesses:**

* While it's valuable to have an **empirical validation**, I felt that section 6 is not as well described and discussed as the rest of the paper. E.g., even the metrics mentioned in the caption of Fig 1 need clarification.

* There are some places where it would be valuable to state which **design decisions / assumptions / choices** play an important role. E.g., it seems to me that Prop 4.1 (that whatever NoPE-AHAT can compute is expressible semi-algebraically) relies on the assumption of ReLU as the non-linearity; I suspect it will also work with any polynomial non-linearity, but not with sigmoid, inverse-tangent, or other choices that have been used in practice. (And this is fine, I just think it's better if the authors can call it out.)

* The paper can also use a clearer treatment of the **datatype** assumed for the transformer model. Section 2.1 starts with *real* vectors, which clearly isn't realistic, at least for arbitrary precision. Later, the ReLU paragraph switches to *rationals*, leaving some lack of clarity. The choice of datatype -- and importantly the *precision* (how many bits are allocated, and whether they depend on the input length $n$ -- is an important consideration in transformer expressivity results, but seems to be lacking here. E.g., when, in the proof of Prop 3.1 (that semi-algebraic counts can be expressed by NoPE-AHAT[U]) the transformer is computing $u_p[j] \in [0,1]$, this is presumably expressed as a rational where the numerator (and denominator) can grow very quickly. Their size should be bounded in terms of $\ell$ and $n$, though. It would be helpful to get some clarity on this front.

* Some of the proof ideas can use a short discussion of the **intuition**. E.g., in the proof of Prop 3.1, we are multiplying two integers (represented as rationals). A priori, it's unclear how a transformer might be able to do so! The key observation, I believe, is that one of the items being multiplied, namely $x_i$, is always a *count* of letters in the input word, and thus distributed across the input as captured by $u_p[i]$ being 0 or 1 at various positions $p$. Thus, one can get around the usual difficulty of multiplication by instead multiplying $y_i$ with a 0 or a 1 at each position, ensuring that there are exactly $x_i$ positions with a 1, and then using attention to aggregate. Is this the right intuition? In any case, adding some intuition would help.

**Questions:**

Please see the comments above about the (implicit) assumptions on the datatype and precision, and on intuitions. Clarity on any of these fronts would be great to have!

---

> ### Author Response · Authors · 2025-11-20
>
> Thanks for your thoughtful review! Below we address your concerns in detail.
>
> ---
>
> ### Comments on "Weaknesses"
>
> > While it's valuable to have an empirical validation, I felt that section 6 is not as well described and discussed as the rest of the paper. E.g., even the metrics mentioned in the caption of Fig 1 need clarification.
>
> We acknowledge that, given our submission is predominantly theoretical, the empirical validations in Section 6 might seem a bit less extensive, but we feel this is mainly due to their complementary nature. To address your concern, we've adjusted our experimental section as follows:
> - We have clarified the description of the figure, providing the necessary information more clearly in the figure description. It should now be clear what the table values mean.
> - We have extended the experimental description, which previously was in Appendix D, and moved it to the main body of the paper (making use of the additional page) to ensure it is self-contained. This also explains the metric BCEWithLogitsLoss of the pytorch-package in more detail.
> - We added some extension of experiments on another class of formal languages with inherent polynomial counting properties and report on overall similar results. For this see the new Appendix D.
>
>
> > There are some places where it would be valuable to state which design decisions / assumptions / choices play an important role. E.g., it seems to me that Prop 4.1 (that whatever NoPE-AHAT can compute is expressible semi-algebraically) relies on the assumption of ReLU as the non-linearity; I suspect it will also work with any polynomial non-linearity, but not with sigmoid, inverse-tangent, or other choices that have been used in practice. (And this is fine, I just think it's better if the authors can call it out.)
>
> Yes, this is correct. All theoretical studies of transformers expressivity have so far only used ReLU as the non-linearity for transformers (as was done in "Attention is all you need" paper). In practice, other smoother approximations of ReLU have also been used including GELU (Gaussian Error Linear Units) and SwiGLU (Swish and GLU), which use exponential functions in some way. We have added this remark in Section 2.
>
> > The paper can also use a clearer treatment of the datatype assumed for the transformer model. [...] The choice of datatype -- and importantly the precision (how many bits are allocated, and whether they depend on the input length $n$ -- is an important consideration in transformer expressivity results, but seems to be lacking here. E.g., when, in the proof of Prop 3.1 (that semi-algebraic counts can be expressed by NoPE-AHAT[U]) the transformer is computing $u_p[j] \in [0,1]$, this is presumably expressed as a rational where the numerator (and denominator) can grow very quickly. Their size should be bounded in terms of $l$ and $n$, though.
>
> This is a good point and was also raised by other reviewers, Prop 3.1 works with rational numbers whose numerators and denominators have $O(\log n)$ bits. We have added this in the new version (see Remark before Section 2.2).
>
>
>
> > Some of the proof ideas can use a short discussion of the intuition. E.g., in the proof of Prop 3.1, we are multiplying two integers (represented as rationals). A priori, it's unclear how a transformer might be able to do so! The key observation, I believe, is that one of the items being multiplied, namely $x_i$, is always a count of letters in the input word, and thus distributed across the input as captured by $u_p[i]$ being 0 or 1 at various positions $p$. Thus, one can get around the usual difficulty of multiplication by instead multiplying $y_i$ with a 0 or a 1 at each position, ensuring that there are exactly $x_i$ positions with a 1, and then using attention to aggregate. Is this the right intuition?
>
> This is exactly right. An additional difficulty comes from the fact that each averaging step introduces a factor $(n+1)$ in the denominator, which is not part of the polynomial to be evaluated. To mitigate this, we first ensure that the polynomial is homogeneous. This implies that every monomial will receive the same number $d$ (=the degree of the polynomial) of factors $(n+1)$. Then, since every monomial is divided by the same factor $(n+1)^d$, the resulting number is non-negative if and only if the original polynomial is non-negative.
>
> Thank you for this suggestion, we have added a description of the overall idea in Section 3.

---

### Author Response · Authors · 2025-11-20
**General Response (1/2)**

We are grateful to all reviewers for raising these important questions, we believe they are important to improve the presentation. We will definitely add these clarifications to the revised version.


> Question: Are there practical NLP tasks that might require nonlinear counting?

Thank you for this question. We have added our clarification below in Section 7.

Since (due to our results) transformers are able to perform any polynomial counting, it follows that they can also approximate any continuous function of the number of occurrences of tokens (the set of polynomials is the universal approximator by the Weierstrass theorem). This might be useful in practical NLP tasks that require computation of nonlinear statistics in the word frequencies. In the paper, we have mentioned its relationship to the so-called *Vector Space Model (VSM)* for text classification and similarity analysis (a classic topic in information retrieval), where the standard method has been to employ Support Vector Machines (SVM), together with kernel analysis (e.g. using polynomial kernels). Our results imply that transformers are expressive enough to perform such tasks (in fact, we showed the added expressivity). We further clarify it here and added this to the new version.

In VSM (see Chapter 10 of the cited book by Shawe-Taylor and Cristianini (2004)), certain structures are dropped; most commonly, the word ordering - that is we obtain a bag of words. That is, a document $D$ is abstracted into a vector $v_D$ indexed by "terms" (could be words, sentences, etc.) that may occur in $D$. We assume that the set $T$ of terms is finite. That is, $v_D[t]$ is a count on the number of occurrences of $t$ in $D$. If we want to compare similarity between two documents $D,D'$, we may consider the Euclidean distance between $v_D$ and $v_{D'}$, which requires a polynomial. Another non-linear distance that is commonly used in this setting is the cosine distance, which can be well approximated by polynomials. In addition, there are often challenges including "related terms" (e.g. husband, wife, and spouse), which are missed when we only use the aforementioned metrics. For this reason, a similarity measure is often learned (see Section 10.2.2 in the cited book by Shawe-Taylor and Cristianini (2004), where SVM is used in combination with kernel analysis like polynomial kernels). Our results essentially show that transformers can solve such a task of identifying text similarity.

A related task is the problem of determining proximity to a human written text, as dictated by the famous Zipf law stating that the frequency of the $k$-th most frequent word is proportional to $1/k$ in a natural language. Following the VSM approach above, we compare using Euclidean distance (or cosine distance) a document $D$ with a predetermined Zipf-vector. This is in general a polynomial expression, and our results show that this can also be captured by transformers.


> Question: Usage of unbounded precision is unrealistic.

On inputs of length $n$, we require just $O(\log n)$ bits of precision. Curiously, to see that, one does not even look at the construction because this is an inherent feature of the NoPE-AHAT model. Namely, numbers in every NoPE-AHAT transformer on inputs of length n can be represented as fractions of $O(\log n)$-bit integers.

This is because every number can be computed in $O(1)$ additions, multiplications and divisions, starting from $x_1, …, x_m \in \\{0, 1, …, n\\}$ – numbers of occurrences of the letters of our alphabet $\Sigma = \\{a_1, ..., a_m\\}$ in a given $n$-length word. This follows from an observation, made in the proof of Proposition 4.1 – after any number of layers, positions with the same input letter have the same value. That is, as in the proof of Proposition 4.1, the values of the transformer after attention layers are given by $m + 1 = O(1)$ numbers $u_{l,0}, …, u_{l,m}$ (note that $m$ is the size of the alphabet and hence is constant). To compute numbers on the next attention layer, we need to compute averages of values for positions where the attention score is maximized. In any position on the next level, it can be maximized on some subset $S \subseteq \\{0, 1, .., m\\}$. Then the corresponding average can be computed as the following expression, requiring just $O(1)$ arithmetic operations (for that, note again that we are summing over a set $S$ of size at most $m +1 = O(1)$):

$(\sum_{i \in S} x_i \cdot  u_{l,i}) / (\sum_{i \in S} x_i )$

After that, to compute the final values in the next layer, in every position we need to perform just a constant number of arithmetic operations of our fixed ReLU network for that layer. Moreover, all expressions needed to evaluate attention scores are quadratic polynomials in the numbers of the respective layer; hence they also just require denominators with O(log n) bits.

---

### Author Response · Authors · 2025-11-20
**General Response (2/2)**

> Question: What is the size of our model in Proposition 3.1?

First, we want to note that all models we are studying in our paper have only one attention head. We added a remark to the definition in Section 2.1.

The embedding dimension and the number of layers of our transformer in Proposition 3.1 depends on the degree $d$ of the polynomial $p$ from the input and the number $M$ of monomials in $p$.
First, it is easy to see that we need at most $O(d)$ layers – essentially, each layer to increase the degree of monomials that we have already computed by 1. In a fine-grained analysis in the appendix, we showed that polynomials of degree $d$ are accepted by NoPE-AHAT[U] with at most $d$ attention layers (see Proposition A.1).

In turn, the embedding dimension in our construction can be bounded by $O(dM)$, – we store the value of each monomial that we need to compute in a separate dimension, we need to compute $M$ monomials of $p$, but computing a monomial $x_{i_1}... x_{i_d}$ takes computing $d$ ``sub-monomial’’, so the dimension bound is multiplied by $d$.

We also added a remark at the end of Section 3 about the size of the constructed AHAT.

---

### Author Response · Authors · 2025-11-28

Dear AC and Reviewers,

We thank you for your valuable input. Owing to the current circumstances with OpenReview, we thought it would be useful to summarize the current state of the rebuttal. We initially received the scores 8, 8, 8, 4, 4. After the rebuttal, the scores are currently 8, 8, 8, 8, 6. We believe that these reflect our satisfactory response to the reviewers' comments.

Once again, we are extremely grateful for all your feedback and the rebuttal, which has improved the paper. We will be happy to answer any additional questions.

Best wishes,

The Authors

---

### Meta-Review · Area_Chair_yCjy · 2026-01-14

**Summary:**

This paper provides a characterization of counting expressivity of transformers. There was some discrepancy in the initial reviews, but the concerns were mainly about the practical applications. The author rebuttals were detailed and were convincing. The paper has clear and important theoretical contributions.

**Reviewer Concerns:**

There were some concerns regarding practical applications.

**Reviewer Scores:**

Both negative reviewers had indicated that they would have increased their scores in response to the author rebuttals.

---

### Decision · Program_Chairs · 2026-01-26

Accept (Poster)